# FLAML version 2.3.3 model-based assessment of gross primary productivity at forest, grassland, and cropland ecosystem sites

Jie Lai [a, b], Yuan Zhang [a], Anzhi Wang [a], Wenli Fei [a], Yiwei Diao [c], Rongping Li [d], Jiabin Wu [a, *]

[a] CAS Key Laboratory of Forest Ecology and Silviculture, Institute of Applied Ecology, Chinese Academy of Sciences, Shenyang 110016, China

[b] University of Chinese Academy of Sciences, Beijing 101408, China

[c] Key Laboratory of Ecosystem Carbon Source and Sink, China Meteorological Administration (ECSS-CMA), Wuxi University, Wuxi, 214105, China

[d] Institute of Atmospheric Environment, China Meteorological Administration, Shenyang 110016, China

* Correspondence: wujb@iae.ac.cn

## Abstract

Accurately estimating Gross Primary Productivity (GPP) in terrestrial ecosystems is essential for understanding the global carbon cycle. Satellite-based Light Use Efficiency (LUE) models are commonly employed for simulating GPP. However, the variables and algorithms related to environmental limiting factors differ significantly across various LUE models, leading to high uncertainty in GPP estimation. In this work, we developed a series of FLAML-LUE models with different variable combinations. These models utilize the Fast Lightweight Automated Machine Learning (FLAML) framework, using variables of LUE models, to investigate the potential of estimating site-scale GPP. Incorporating meteorological data, eddy covariance measurements, and remote sensing indices, we employed FLAML-LUE models to assess the impact of various variable combinations on GPP across different temporal scales, including daily, 8-day, 16-day, and monthly intervals. Cross-validation analyses indicated that the FLAML-LUE model performs excellently in GPP prediction, accurately simulating

both its temporal variations and magnitude, particularly in mixed forests and coniferous forests, with average $R^2$ values for daily-scale simulations reaching 0.92 and 0.91, respectively. However, the model performed less effectively in alpine shrubland and typical grassland ecosystems, though it still outperformed both MODIS GPP and PML GPP in terms of performance. Furthermore, the model's adaptability under extreme climate conditions was evaluated, and the results showed that high temperatures and high VPD lead to a slight decrease in model accuracy, though $R^2$ remains around 0.8. Under drought conditions, the model's performance improved slightly in croplands and evergreen broadleaf forests, although it declined at some sites. This study offers an approach to estimate GPP fluxes and evaluate the impact of variables on GPP estimation. It has the potential to be applied in predicting GPP for different vegetation types at a regional scale.

**Keywords:** Light Use Efficiency; Gross Primary Productivity; Automated Machine Learning; Fast Lightweight Automated Machine Learning

## 1. Introduction

The global carbon budget mainly addresses the carbon reserves in the atmosphere, oceans, and terrestrial (Barbour, 2021), with terrestrial ecosystems being vital for regulating the global carbon cycle (Gherardi and Sala, 2020; Landry and Matthews, 2016). Terrestrial ecosystems primarily absorb atmospheric carbon dioxide through the process of plant photosynthesis, which is crucial for regulating climate and mitigating global warming (Sellers et al., 2018; Beer et al., 2010; Cox et al., 2000). Gross primary productivity (GPP) is a critical measure of carbon exchange between terrestrial

ecosystems and the atmosphere (Menefee et al., 2023). Accurate quantification of GPP

is essential for evaluating carbon balance and comprehending the response of terrestrial

ecosystems to climate change (Sellers et al., 2018).

The primary method currently used for measuring $CO_2$ exchange between

ecosystems and the atmosphere is the eddy covariance technique (Chen et al., 2020; Yu

et al., 2016). This technique precisely measures Net Ecosystem Exchange (NEE), which

is the difference between the carbon released by ecosystem respiration (ER) and the

carbon taken up by photosynthesis (Bhattacharyya et al., 2013). While flux observation

sites based on the eddy covariance (EC) technique can dynamically monitor site-scale

carbon fluxes, expanding their findings to larger regional scales remains challenging,

mainly due to the sparse and spatially non-uniform distribution of flux sites (Xie et al.,

2023; Jung et al., 2020). Remote sensing data is widely used in ecosystem carbon cycle

research as it can provide information on the spatial dynamics of vegetation and climate

at a larger scale (Xiao et al., 2019). By extrapolating spatially using models that

incorporate remote sensing and climate data, it is possible to estimate global GPP based

on observations of GPP at the site level. Therefore, remote sensing has become a crucial

data resource for estimating GPP (Cai et al., 2021; Xiao et al., 2019; Wang et al., 2011).

Light Use Efficiency (LUE) models based on satellite observations are commonly

employed to simulate GPP (Zhang et al., 2023; Zhang et al., 2015; Jiang et al., 2014).

Such models include Physiological Principles Predicting Growth using Satellite data

(3-PGS, Coops and Waring, 2001), the Carnegie-Ames- Stanford Approach Model

(CASA, Potter et al., 1993), the Eddy Covariance–Light Use Efficiency Model (EC-

LUE, Yuan et al., 2010, 2007), the MODIS Global Terrestrial Gross and Net Primary
Production (MOD17, Running et al., 2004), the Vegetation Photosynthesis Model
(VPM, Xiao et al., 2003), and the Vegetation Photosynthesis and Respiration Model
(VPRM, Mahadevan et al., 2008). Among all the forecasting methods (Coops and
Waring, 2001; Potter et al., 1993), the LUE model is widely utilized for simulating the
spatio-temporal dynamics of GPP due to its simplicity and strong theoretical foundation.
Over the past few decades, numerous GPP models utilizing LUE have been developed
(Pei et al., 2022).
Despite significant advances in LUE theory for GPP estimation, uncertainties
persist in GPP models utilizing LUE. Firstly, differences in environmental limiting
factors among various LUE models contribute significantly to the uncertainty in GPP
estimation. For example, Cai et al. (2014) found a strong positive correlation between
water effectiveness and GPP estimate factors, while other studies found that the LUE
model estimates of GPP were strongly correlated with the vegetation index, which
affects the photosynthetic capacity of vegetation through leaf nitrogen content
(Peltoniemi et al., 2012; Ercoli, 1993).
Recently, with the massive accumulation of satellite data and ground-based
observations, more and more studies have applied machine learning (ML) methods to
model ecosystem processes (Zhao et al., 2019; Alemohammad et al., 2017; Chaney et
al., 2016). ML is a modeling solution that differs from simple regression models and
complex simulation models in its approach. It is very effective in handling large-scale
multivariate data with complex relationships between predictors (Reichstein et al., 2019;
Tramontana et al., 2016). These data-driven models are particularly suited for capturing
nonlinear ecosystem dynamics but often require large training datasets and may lack
explicit links to real-world processes. However, their ability to uncover spatial patterns
without process-based constraints makes them valuable for spatial predictions.
Consequently, ML-based approaches have gained popularity in recent years. For
example, Kong et al. (2023) developed a hybrid model that combines ML and LUE
model to estimate GPP. This hybrid model improves the LUE model by integrating a
machine learning approach (MLP, multi-layer perceptron), and estimates GPP using the
MLP-based LUE framework along with additional required inputs. Chang et al. (2023)
constructed RFR-LUE models that utilize the Random Forest Regression (RFR)
algorithm with variables of LUE models to assess the potential of site-scale GPP
estimation.
Lately, Automated Machine Learning (AutoML) has demonstrated significant
potential in constructing data-driven models automatically (Zheng et al., 2023).
Numerous sophisticated open-source AutoML frameworks have been suggested by
computer scientists, including Automated WEKA (Auto-WEKA, Thornton et al., 2013),
H2O AutoML (H2O, LeDell and Poirier, 2020), Tree-based Pipeline Optimization Tool
(TPOT, Melanie, 2023), Automated Machine Learning with Gluon (AutoGluon,
Erickson et al., 2020), Fast Lightweight Automated Machine Learning (FLAML, Wang
et al., 2021), and AutoKeras (Rosebrock, 2019). These frameworks are extensively used
in finance, manufacturing, healthcare, and mobile communications, among other fields
(Adams et al., 2020), with FLAML being particularly favored for its efficiency in rapid

prototyping and deployment in research and production settings. FLAML is a powerful framework for AutoML, known for its speed in identifying top-performing models and optimal hyperparameters through parallel optimization and smart search algorithms. FLAML integrates several effective search strategies, outperforming other leading AutoML libraries on large benchmarks even with constrained budgets (Wang et al., 2021).

In this research, a new model called FLAML-LUE was created by combining FLAML model with LUE-based models, the latter provides the key variables of vegetation growth for modeling. Such knowledge-and-data-driven models aim to reduce the large uncertainty in estimating GPP. The specific objectives of this study are: (1) to evaluate the overall performance of models using different input variables, including the fraction of photosynthetically active radiation absorbed by vegetation (fPAR) and various water stress indicators, across multiple sites and vegetation types based on eddy covariance observations; (2) to assess model performance under extreme climatic conditions, such as high temperature, elevated vapor pressure deficit (VPD), and drought.

## 2. Materials and methods

## 2.1 Site description

**Figure 1** displays the geographical locations of the 20 flux sites selected for the study. These sites are situated in various climatic zones and ecosystem types including forest, grassland, and cropland. The observation data for these sites comes from the Science Data Bank (SDB, https://www.scidb.cn/en/). Detailed information about the

sites is provided in **Table 1**.

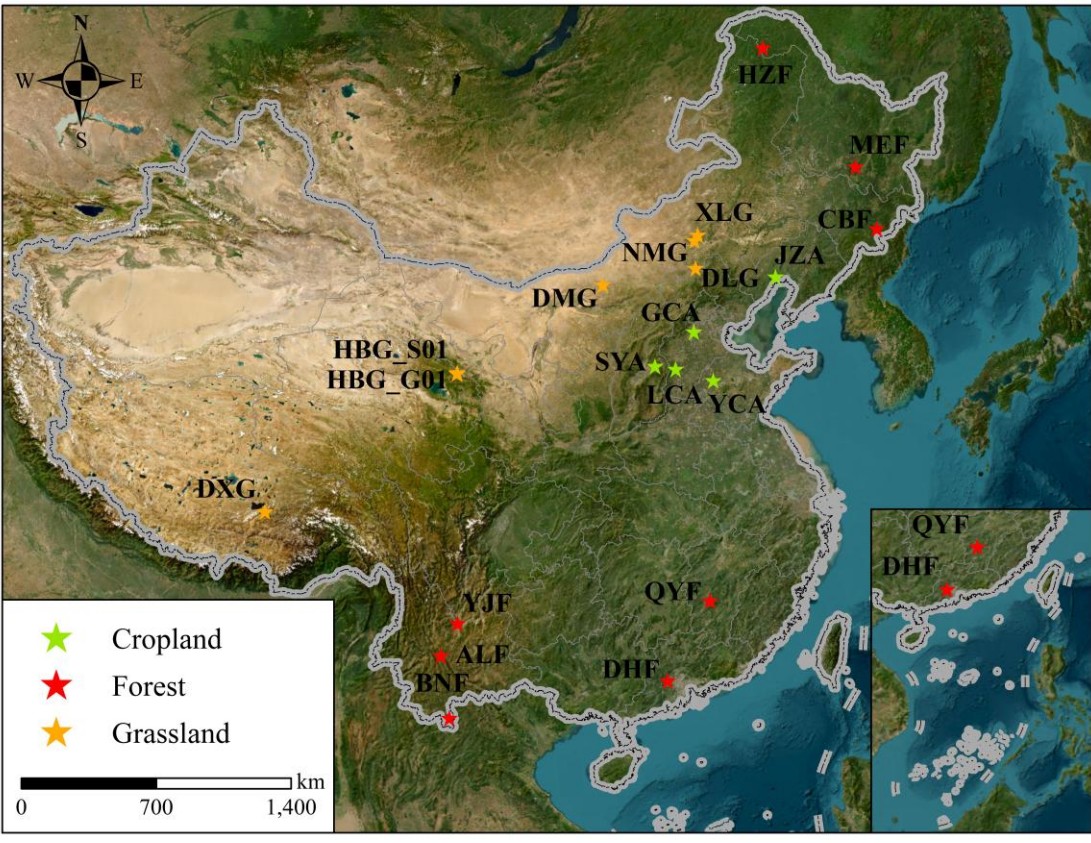


**Figure 1.** The location map of the flux site is based on the map approved by the National Surveying
and Mapping Bureau of China (Approval No. GS (2019)1822). The topographic map is derived
from data provided by Esri, Maxar, Earthstar Geographics, and the GIS User Community (Service
Layer Credits).
**Table 1**
Basic information on the 20 flux stations.

| Site | Longitude (°E) | Latitude (°N) | Ecosystem type | Time Range | Classified |
|------|----------------|---------------|----------------|------------|------------|
| HZF | 123.018 | 51.781 | Forest | 2014-2018 | NF |
| MEF | 127.668 | 45.417 | Forest | 2016-2018 | DBF |
| CBF | 128.096 | 42.403 | Forest | 2003-2010 | MF |
| QYF | 115.058 | 26.741 | Forest | 2003-2010 | NF |
| ALF | 101.028 | 24.541 | Forest | 2009-2013 | EBF |
| DHF | 112.534 | 23.173 | Forest | 2003-2010 | MF |
| BNF | 101.577 | 21.614 | Forest | 2003-2015 | EBF |
| YJF | 101.827 | 26.080 | Forest | 2013-2015 | SAV |
| XLG | 116.671 | 43.554 | Grassland | 2006-2014 | GRA |
| NMG | 116.404 | 43.326 | Grassland | 2003-2010 | Grassland |
| DLG | 116.284 | 42.047 | Grassland | 2006-2015 | Grassland |
| DMG | 110.328 | 41.644 | Grassland | 2015-2018 | Grassland |

| | | | | | |
|---|---|---|---|---|---|
| HBG_G01 | 101.313 | 37.613 | Grassland | 2015-2020 | MEA |
| HBG_S01 | 101.331 | 37.665 | Grassland | 2003-2013 | SHR |
| DXG | 91.066 | 30.497 | Grassland | 2003-2010 | MEA |
| JZA | 121.202 | 41.148 | Cropland | 2005-2014 | SC |
| GCA | 115.735 | 39.149 | Cropland | 2020-2022 | DC |
| SYA | 113.200 | 37.750 | Cropland | 2012-2014 | SC |
| LCA | 114.413 | 37.531 | Cropland | 2013-2017 | DC |
| YCA | 116.570 | 36.829 | Cropland | 2003-2010 | DC |

Note: Vegetation types in the table are classified based on the land cover characteristics of each flux
site and are used in subsequent model simulations. NF: Needle-leaved Forest; DBF: Deciduous
Broadleaved Forest; MF: Mixed Forest; EBF: Evergreen Broadleaved Forest; SAV: Savannas; GRA:
Typical Grassland; MEA: Alpine Meadow; SHR: Shrubs; SC: Single Cropping; DC: Double
Cropping.

## 2.2 Data

## 2.2.1 Eddy covariance data

EC data were collected at 20 sites, including 8 forests sites, 7 grasslands sites, and
5 cropland sites (**Table 1**). Flux and meteorological data were collected every half hour
from the mentioned sites. The flux and meteorological data underwent standardized
quality control and corrections, ensuring high reliability and making them suitable for
validating various GPP models and remote sensing observations. However, ER data
were missing at some sites (DLG, LCA, XLG). To address this, the Lloyd & Taylor
equation (Reichstein et al., 2005; Lloyd and Taylor, 1994) was applied to estimate ER
based on nocturnal respiration data. Daytime and nighttime periods were distinguished
using shortwave radiation (Rg), with a threshold of 10 W/m². The temperature–response
relationship derived from nighttime ER was extrapolated to estimate daytime ER. This
is a commonly used method for processing flux data at flux tower sites.

$$R_{eco} = R_{eco.ref} \exp \left( E_0 \left( \frac{1}{T_{ref}-T_0} - \frac{1}{T_{air}-T_0} \right) \right) \tag{1}$$

$$GPP = ER - NEE \tag{2}$$

In equation (1), $R_{eco}$ is the nocturnal ecosystem respiration value, $R_{eco.ref}$ is the ER value
at the reference temperature, $T_{ref}$ is the reference temperature (298.16K), $E_0$ is constant
(308.56K), $T_0$ is the minimum temperature at which respiration stops, set at 227.13K,
and $T_{air}$ is the air temperature or soil temperature (K). Daytime GPP was then estimated
by subtracting NEE from the total daytime ER.

**2.2.2 MODIS data**


In this study, remote sensing data were primarily obtained from the Moderate
Resolution Imaging Spectroradiometer (MODIS). MODIS data offer a spatial
resolution of 500 meters and an 8-day temporal resolution. These datasets were sourced
from the Google Earth Engine (GEE) platform (Gorelick et al., 2017). To align with the
spatial and temporal scales of flux tower observations and reduce the impact of missing
data (Schmid, 2002), we applied the Savitzky-Golay smoothing filter with a window
size of 10 to process the vegetation indices. Vegetation and water indices derived from
MODIS data included the enhanced vegetation index (EVI), normalized difference
vegetation index (NDVI), and land surface water index (LSWI), which were calculated
using the formulas presented in **Table 2**.

**2.2.3 ERA5-LAND**


ERA5-Land (Hersbach et al., 2020) is a global high-resolution reanalysis dataset
produced by the European Centre for Medium-Range Weather Forecasts (ECMWF)
under the Copernicus Climate Change Service (C3S). It provides hourly land surface
variables at a spatial resolution of 0.1°, generated using a dedicated land surface model
driven by the ERA5 climate reanalysis. The dataset integrates advanced land surface
modeling and data assimilation techniques, offering a wide range of variables such as
air temperature, soil moisture, precipitation, and snow depth. In this study, site-specific
variables including air temperature (T), soil water content (SW), precipitation (Pre),
and leaf area index (LAI) were extracted from ERA5-Land. In addition,
photosynthetically active radiation (PAR), evapotranspiration fraction (EF), VPD and
relative humidity (RH) were calculated and derived from available ERA5-Land
variables using GEE.

### 2.2.4 SPEI Database, Version 2.10

The SPEI Database, Version 2.10 (Vicente-Serrano et al., 2010) provides global
data of the Standardized Precipitation-Evapotranspiration Index (SPEI) across temporal
scales from 1 to 48 months. Developed by the Climatic Research Unit (CRU), this
dataset combines precipitation and potential evapotranspiration (PET) to assess drought
conditions. Negative SPEI values indicate drought, while positive values signify wet
periods. In this study, SPEI values less than -1.5 were used to identify drought months
at each flux station, highlighting significant moisture deficits that affect vegetation
growth and ecosystem productivity (Qian et al., 2024).

### 2.3 Model Construction

Most LUE models typically incorporate four main groups of variables: PAR, fPAR,
temperature, and water-related stress indicators. In previous studies, vegetation indices
such as EVI, NDVI, or LAI have been widely used as proxies for fPAR, representing
the fraction of PAR absorbed by the plant canopy (Chang et al., 2023; Qian et al., 2024).
In this study, we selected six water-related indicators based on their ecological
relevance: plant-based indicators (LSWI and EF), soil-based indicators (SW), and
atmospheric indicators (VPD, precipitation, and relative humidity). Previous research
has shown that plant-based indicators like LSWI and EF effectively capture canopy-
level drought stress (Anderson et al., 2007; Xiao et al., 2004). Soil moisture regulates
water availability at the root level, which strongly influences photosynthetic activity,
particularly under water-limited conditions (Vicca et al., 2014; Reichstein et al., 2007).
Meanwhile, atmospheric indicators such as VPD, precipitation, and RH influence
stomatal conductance and transpiration by altering the vapor pressure gradient between
the leaf surface and the surrounding air (Wang et al., 2018; Novick et al., 2016). To
assess the relative importance of these different types of water stress indicators in
estimating GPP, we developed machine learning models using each group individually.
This allowed us to identify the most effective type of water-related variable for
simulating GPP across diverse ecosystems within the LUE modeling framework.
The flowchart of this study is shown in **Figure 2**.

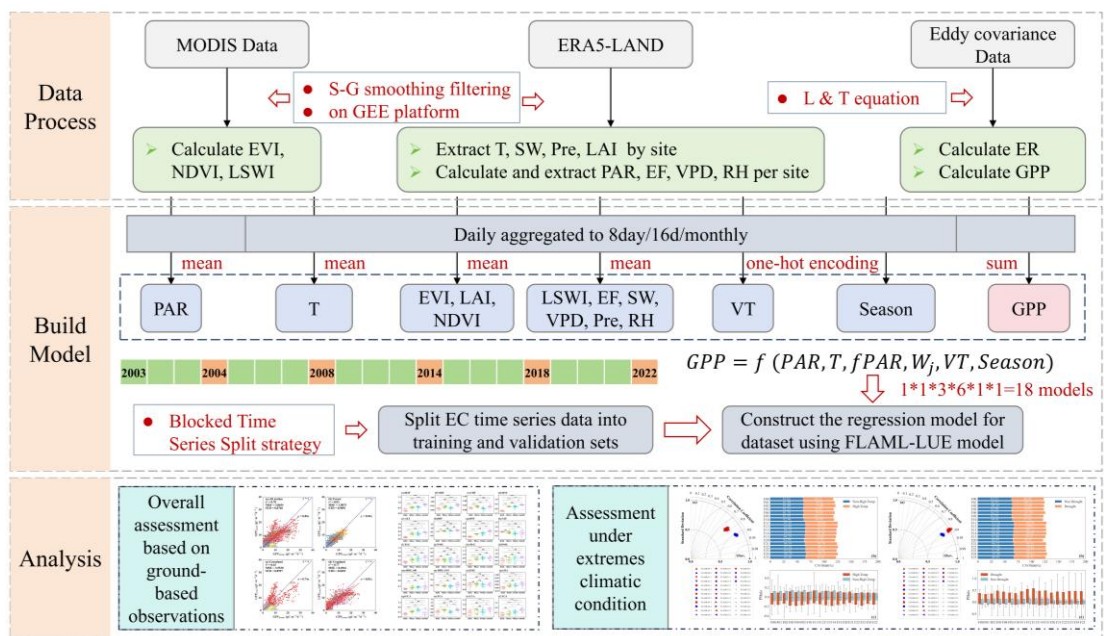


**Figure 2.** Flowchart of this study. S-G smoothing filtering: Savitzky-Golay smoothing filtering method, L & T equation: Lloyd & Taylor equation.

## 2.3.1 Data pre-processing and splitting strategy

The primary datasets for estimating GPP with FLAML-LUE models include multi-year continuous EC flux data, satellite-based observations, and ERA5-Land climate reanalysis data. Prior research (Jung et al., 2011) has demonstrated notable seasonal fluctuations in GPP, we divided the time series data into four distinct seasons. Moreover, the vegetation cover type, which varies across different ecosystems, greatly impacts the accuracy of GPP simulation (Chang et al., 2023). Hence, we integrate vegetation type as a factor in our model.

The pre-processed dataset was divided into training and testing sets using the Blocked Time Series Split strategy. Given the temporal dependency of the data, standard cross-validation is not suitable for time series analysis (Reichstein et al., 2019). Instead, a block-based and non-continuous split is applied to preserve the temporal structure. In this approach, the time series is partitioned into several non-overlapping continuous training blocks (e.g., 2003-2005, 2007-2009, 2011-2013, 2015-2017, 2019-2021), with independent years reserved as the validation set following each training block (e.g., 2006, 2010, 2014, 2018, 2022). This strategy ensures that the temporal order is maintained, preventing future data from leaking into the training process and thus avoiding invalid predictions. Additionally, the method incorporates validation over multiple periods, enabling the assessment of model generalization across different climate conditions, which is crucial for evaluating the model's robustness under varying environmental scenarios.

**Table 2**

Predictor variables for driving the FLAML models and their specifications.

| | Variable | Acquired method (formula) | Original Spatial Resolution | Data Source |
|---|---|---|---|---|
| fPAR | EVI | $2.5 \times (R_{nir} - R_{red})/ (R_{nir} + 6.0 \times R_{red} - 7.5 \times R_{blue} + 1)$ | 500m | MOD09GA |
| | NDVI | $(R_{nir} - R_{red})/ (R_{nir} + R_{red})$ | | |
| | LAI | - | ~10km | ERA5-Land |
| Water | LSWI | $(R_{nir} - R_{swir})/ (R_{nir} + R_{swir})$ | 500m | MOD09GA |
| | EF (%) | $EF = LE/(LE+H)$ | ~10km | ERA5-Land |
| | SW ($m^3/ m^3$) | - | ~10km | ERA5-Land |
| | VPD | $VPD = e_s - e$ $e = 6.112 \times exp(( 17.67 \times T_d) + (243.5 + T_d))$ $e_s = 6.112 \times exp(( 17.67 \times T) + (243.5 + T))$ | ~10km | ERA5-Land |
| | Pre (mm) | - | ~10km | ERA5-Land |
| | RH (%) | $RH = (e/e_s) \times 100$ | ~10km | ERA5-Land |
| Radiation | PAR ($\mu$ mol $m^{-2}$ $s^{-1}$) | - | ~10km | ERA5-Land |
| Temperature | T (℃) | - | ~10km | ERA5-Land |
| VT | EBF, DBF, NF, MF, GRA, MEA, SHR, SC, DC | One-hot encoding | invariant | - |
| Season | Spring, Summer, Autumn, Winter | One-hot encoding | invariant | - |

Note: EVI: Enhanced Vegetation Index, NDVI: Normalized Difference Vegetation Index, LAI: Leaf Area Index, LSWI: Land Surface Water Index, EF: Evaporative Fraction, SW: Surface Soil Moisture, VPD: Vapor Pressure Deficit, Pre: Precipitation, RH: Relative Humidity, PAR: Photosynthetically Active Radiation, and T: Air Temperature. NF: Needle-leaved Forest; DBF: Deciduous Broadleaved Forest; MF: Mixed Forest; EBF: Evergreen Broadleaved Forest; SAV: Savannas; GRA: Typical Grassland; MEA: Alpine Meadow; SHR: Shrubs; SC: Single Cropping; DC: Double Cropping. In the formulas for EVI and NDVI, $R_{nir}$, $R_{red}$, $R_{blue}$, $R_{swir}$ represent the surface reflectance in the near-infrared (NIR), red, and blue spectral bands, respectively. In the EF calculation formula, LE refers to latent heat flux, while H represents sensible heat flux. In the RH formula, e is the actual vapor pressure, $e_s$ is the saturation vapor pressure, $T_d$ is the dew point temperature, and T is the air temperature.

## 2.3.2 Automated Machine Learning (AutoML)

Instead of applying a specific ML method like RF for building regression models, we utilize the lightweight Python library "FLAML" version 2.3.3 (Wang et al., 2021) for the AutoML task (Metin and Bilgin, 2024). FLAML optimizes the search process by balancing computational cost and model error, iteratively selecting the learner, hyperparameters, sample size, and resampling strategy (Wang et al., 2021).

For our regression tasks, AutoML was configured with the "auto" option for the
estimator list, focusing on optimizing the $R^2$ metric and using a time budget of 120
seconds per run. Under this "auto" setting, FLAML explores a variety of built-in
regression estimators, including:
1.  LightGBM (Ke et al., 2017): a histogram-based gradient boosting method designed

for speed and scalability;

2.  XGBoost (Chen and Guestrin, 2016): a regularized gradient boosting framework

known for its robustness and accuracy;

3.  CatBoost (Prokhorenkova et al., 2018): efficiently handles categorical features and

reduces overfitting via ordered boosting;

4.  Random Forest (Breiman, 2001): an ensemble method utilizing bootstrap

aggregation of decision trees;

5.  Extra Trees (Geurts et al., 2006): enhances randomness in split point selection for

tree construction;

6.  Histogram-based Gradient Boosting (Brownlee, 2020), accelerate training through

feature binning;

7.  K-Nearest Neighbors (Cover and Hart, 1967): a non-parametric distance-based

algorithm relying on local data density;

8.  Transformer models (Vaswani et al., 2023), deep learning architectures leveraging

self-attention mechanisms, adapted here for structured data regression.

Collectively, these estimators span a broad algorithmic spectrum, including
ensemble learning, distance-based methods, and neural networks, enabling FLAML to
automatically identify the optimal model architecture for the dataset and objective.

### 2.3.3 Model development

Eighteen FLAML-LUE model variations were constructed for all sites by
combining different permutations of six input factor groups, as described in Eq. (3) and
detailed in **Table 3**. Technically, the term "FLAML-LUE" does not refer to a direct
implementation of a mechanistic LUE model. Instead, it reflects a hybrid modeling
strategy, through which we incorporate key explanatory variables that originate from
LUE theory—such as fPAR, light-use efficiency modifiers, and environmental stress
indicators (e.g., VPD, temperature, and water stress indices)—into an automated
machine learning framework (FLAML). These variables capture the main drivers of
vegetation productivity in traditional LUE models. Their integration enables FLAML
to build models that are both ecologically grounded and predictive, effectively
balancing model interpretability and accuracy.

$$GPP = f\ (PAR, T, fPAR, W_j, VT, Season) \tag{3}$$

where, the $fPAR$ include EVI, NDVI, and LAI; $W_j$ denotes moisture factors including
LSWI, EF, SW, PDSI, Pre, RH; $VT$ represents vegetation types, in which forest
ecosystems include: EBF, DBF, NF, MF, and SAV; grassland ecosystems include
GRA, MEA, and SHR, and farmland ecosystems include SC and DC; $Season$
represents the season in which the original data were acquired.
**Table 3**
Input variable combinations of fPAR and water stress indicators.

| Group | Input variables | Group | Input variables | Group | Input variables |
|---|---|---|---|---|---|
| FLAML00 | NDVI, LSWI | FLAML10 | EVI, LSWI | FLAML20 | LAI, LSWI |
| FLAML01 | NDVI, EF | FLAML11 | EVI, EF | FLAML21 | LAI, EF |
| FLAML02 | NDVI, SW | FLAML12 | EVI, SW | FLAML22 | LAI, SW |

| FLAML03 | NDVI, VPD | FLAML13 | EVI, VPD | FLAML23 | LAI, VPD |
| FLAML04 | NDVI, Pre | FLAML14 | EVI, Pre | FLAML24 | LAI, Pre |
| FLAML05 | NDVI, RH | FLAML15 | EVI, RH | FLAML25 | LAI, RH |

Note: EVI: Enhanced Vegetation Index, NDVI: Normalized Difference Vegetation Index, LAI: Leaf Area Index, LSWI: Land Surface Water Index, EF: Evaporative Fraction, SW: Surface Soil Moisture, VPD: Vapor Pressure Deficit, Pre: Precipitation, RH: Relative Humidity.

## 2.3.4 Model performance evaluation methods

To evaluate the simulation accuracy of the FLAML-LUE model in estimating GPP, we employed a suite of widely used statistical metrics to quantify the agreement between modeled and observed values (Qian et al., 2024; Chang et al., 2023; Tramontana et al., 2016). Specifically, we calculated the coefficient of determination ($R^2$), Pearson correlation coefficient (R), normalized unbiased root mean square error (nuRMSE), and normalized standard deviation (NSD, $\hat{\sigma}_f$), based on GPP observations from flux towers and model simulations. The Taylor diagram (Taylor, 2001) was utilized to provide a visual summary of the model's performance, incorporating R, nuRMSE, and NSD.

$$R^2 = \frac{\left[\sum_{t=1}^{T}(f_t - \bar{f})(o_t - \bar{o})\right]^2}{\sum_{t=1}^{T}(f_t - \bar{f})^2 \sum_{t=1}^{T}(o_t - \bar{o})^2} \tag{4}$$

$$R = \frac{\frac{1}{T}\sum_{t=1}^{T}(f_t - \bar{f})(o_t - \bar{o})}{\sigma_f \sigma_o} \tag{5}$$

$$nuRMSE = \frac{uRMSE}{\sigma_o} = \frac{1}{\sigma_o}\sqrt{\frac{1}{T}\sum_{t=1}^{T}\left[(f_t - \bar{f}) - (o_t - \bar{o})\right]^2} \tag{6}$$

$$\hat{\sigma}_f = \frac{\sigma_f}{\sigma_o} = \frac{1}{\sigma_o}\sqrt{\frac{1}{T}\sum_{t=1}^{T}\left((f_t - \bar{f})\right)^2} \tag{7}$$

$$\sigma_o = \sqrt{\frac{1}{T}\sum_{t=1}^{T}\left((o_t - \bar{o})\right)^2} \qquad (8)$$

where, $o_t$ represents the observed GPP from the flux tower, $f_t$ denotes the simulated GPP from FLAML-LUE model, $\bar{o}$ represents the average of observed GPP from the flux tower, $\bar{f}$ represents the average of estimated GPP from the GPP product, $t$ represents the corresponding ID for the GPP data, and n represents the total count of GPP data for the site. $\sigma_o$ represent the standard deviations of the observed GPP. A higher $R^2$ value indicates better consistency between the estimated GPP and the flux GPP.

In addition, the Taylor Skill Score (TSS) was computed to quantitatively assess the overall agreement between simulations and observations, with higher values indicating better performance.

$$TSS = \frac{4(1+R)}{\left(\hat{\sigma}_f + \frac{1}{\hat{\sigma}_f}\right)^2 (1+R_0)} \qquad (9)$$

where $\sigma_f$ represent the standard deviations of the model simulation, and $R_0$ denotes the maximum possible correlation coefficient (in this study, $R_0 = 1$). The TSS ranges from 0 to 1, with a higher TSS indicating better overall model performance relative to the observations.

To further investigate model bias across sites, the percent bias (PBias) was introduced (Qian et al., 2024). Positive PBias values indicate overestimation by the model, while negative values suggest underestimation. The closer the PBias is to zero, the more accurate the model's estimations. The calculation formula is as follows:

$$PBias = \frac{\sum_{t=1}^{T}(f_t - o_t)}{\sum_{t=1}^{T} o_t} \times 100\% \qquad (10)$$

To evaluate the model's ability to capture GPP dynamics under extreme climate
conditions, we identified heatwaves and high VPD events using the 95th percentile of
historical meteorological records (Stefanon et al., 2012; Anderson and Bell, 2010).
Drought events were defined as months with SPEI less than -1.5 (Ayantobo et al., 2019;
Gumus, 2023) . These definitions enabled us to evaluate model performance under
extreme environmental stresses (Qian et al., 2024, 2023).

$$CV_{Rmse} = \frac{\sqrt{\frac{1}{T}\sum_{t=1}^{T}(f_t - o_t)^2}}{\bar{o}} \times 100\% \qquad (11)$$

To determine whether model performance differed significantly across temporal
resolutions (daily, 8-day, 16-day, and monthly), we conducted paired t-tests at a 0.05
significance level. All statistical analyses were performed in Python 3.9 using libraries
including numpy, pandas, scipy, matplotlib, sklearn, and flaml. Complementary
visualizations were produced in R using ggplot2, ggpubr, and readxl.
**3. Results**
**3.1 Overall Model Evaluation Based on Ground-Based Observations**
To evaluate the model performance at the site level, the accuracy of the 18 FLAML-
LUE models was assessed using test datasets from individual flux tower sites. The
algorithms selected by each FLAML-LUE model are listed in **Table S1**. Notably, the
Extra-Trees algorithm was most frequently chosen as the best-performing model. Extra
Trees is an ensemble method that constructs multiple unpruned decision trees and
introduces high randomness in both feature and threshold selection, which enhances
generalization and reduces overfitting, particularly in noisy or high-dimensional
datasets. The consistent selection of Extra Trees suggests that FLAML tends to favor
models with higher stochasticity and ensemble structures under the given data and
computational constraints.
**Figure 3** presents the R, nuRMSE, and NSD values for the 18 models. As shown
in **Figure 3u**, the model performance shows relatively small differences across different
combinations of input indicators. Specifically (**Table 4**), the overall $R^2$ of the different
FLAML-LUE models ranged from 0.78 to 0.82, while nuRMSE values ranged from
0.4240 to 0.4670.
Among the fPAR-related indices, the model driven by EVI performed slightly
better ($R^2 = 0.82$, nuRMSE = 0.4265) than those driven by NDVI ($R^2 = 0.80$, nuRMSE
= 0.4524) and LAI ($R^2 = 0.79$, nuRMSE = 0.4561). Regarding moisture stress indicators,
the model using LSWI as input achieved the best performance ($R^2 = 0.82$, nuRMSE =
0.4298), followed by those using VPD ($R^2 = 0.80$, nuRMSE = 0.4455) and RH ($R^2 =$
0.80, nuRMSE = 0.4450). Models driven by EF ($R^2 = 0.80$, nuRMSE = 0.4487), SW
($R^2 = 0.80$, nuRMSE = 0.4505), and Pre ($R^2 = 0.80$, nuRMSE = 0.4503) performed
slightly worse, though the differences were minimal.
As shown in **Table 5**, the performance of the FLAML-LUE model varies
considerably across different sites, with the average $R^2$ ranging from 0.17 at DXG to
0.92 at CBF and HBG_G01. Notably, this variation was primarily attributed to site-
level differences rather than the combinations of input indicators (**Figure 3**),
highlighting the influence of land cover type and climatic conditions on model
performance.
The best model performance was observed at the HZF, MEF, CBF and HBG_G01
sites ($R^2$ > 0.85, TSS > 0.9), followed by QYF, DLG, JZA, and SYA ($R^2$ > 0.75, TSS >
0.88). Within forest ecosystems, the model performed better in MF, NF, and DBF than
in EBF (ALF, BNF) and savannas (YJF). MF, which include both evergreen conifers
and deciduous broadleaf species, exhibit distinct seasonal variations that can be
effectively captured by satellite imagery. In contrast, EBF show minimal seasonal
greenness variation, leading to larger modeling bias in GPP estimation.
In grassland ecosystems, the model performed better for shrublands and typical
steppe than for alpine meadows (**Tables S4 and S5**). Alpine meadows, characterized
by short growing seasons and harsh high-altitude climates, often experience strong
environmental disturbances and large GPP fluctuations, making them more difficult to
model accurately. In contrast, typical steppe and alpine shrublands display clearer
phenological rhythms and stronger photosynthetic activity, making their GPP dynamics
easier to capture.
In cropland ecosystems, all sites demonstrated relatively strong model performance
($R^2$ > 0.6, TSS > 0.80). Compared to natural grasslands or alpine meadows, croplands
are usually monocultures with stable phenology and simpler canopy structures, which
aid in more accurate GPP modeling.
Notably, at the DXG site, the model achieved a high TSS (0.8326) but a relatively
low $R^2$ (0.17), primarily due to the large performance variation among different index
combinations. As shown in **Table S4**, all six NDVI-driven models (FLAML10-
FLMAL15) have negative R² values, significantly reducing the overall model accuracy
at this site.
**Table 4**
Summary of evaluation metrics for FLAML-LUE model performance across all validation sites.

| FLAML | $R^2$ | R | NSD | nuRMSE | TSS |
|---|---|---|---|---|---|
| **FLAML00** | **0.82** | 0.91 | 0.8806 | 0.4240 | **0.9378** |
| FLAML01 | 0.82 | 0.90 | 0.8717 | 0.4301 | 0.9340 |
| FLAML02 | 0.82 | 0.90 | 0.8810 | 0.4299 | 0.9365 |
| FLAML03 | 0.82 | 0.91 | 0.8748 | 0.4250 | 0.9360 |
| FLAML04 | 0.82 | 0.91 | 0.8763 | 0.4254 | 0.9363 |
| FLAML05 | 0.82 | 0.91 | 0.8691 | 0.4244 | 0.9346 |
| FLAML10 | 0.82 | 0.90 | 0.8638 | 0.4277 | 0.9323 |
| FLAML11 | 0.79 | 0.89 | 0.8641 | 0.4620 | 0.9237 |
| FLAML12 | 0.79 | 0.89 | 0.8686 | 0.4597 | 0.9256 |
| FLAML13 | 0.79 | 0.89 | 0.8592 | 0.4539 | 0.9244 |
| FLAML14 | 0.79 | 0.89 | 0.8629 | 0.4585 | 0.9243 |
| FLAML15 | 0.80 | 0.89 | 0.8671 | 0.4525 | 0.9271 |
| FLAML20 | 0.81 | 0.90 | 0.8610 | 0.4376 | 0.9291 |
| FLAML21 | 0.79 | 0.89 | 0.8551 | 0.4542 | 0.9230 |
| FLAML22 | 0.79 | 0.89 | 0.8597 | 0.4618 | 0.9225 |
| FLAML23 | 0.79 | 0.89 | 0.8562 | 0.4577 | 0.9225 |
| FLAML24 | 0.78 | 0.88 | 0.8543 | **0.4670** | 0.9194 |
| FLAML25 | 0.79 | 0.89 | 0.8590 | 0.4582 | 0.9232 |
| Statistics | | | | | |
| EVI | **0.82** | 0.90 | 0.8756 | 0.4265 | **0.9359** |
| NDVI | 0.80 | 0.89 | 0.8643 | 0.4524 | 0.9262 |
| LAI | 0.79 | 0.89 | 0.8576 | 0.4561 | 0.9233 |
| LSWI | **0.82** | 0.90 | 0.8685 | 0.4298 | **0.9330** |
| EF | 0.80 | 0.89 | 0.8636 | 0.4487 | 0.9269 |
| SW | 0.80 | 0.89 | 0.8698 | 0.4505 | 0.9282 |
| VPD | 0.80 | 0.90 | 0.8634 | 0.4455 | 0.9276 |
| Pre | 0.80 | 0.89 | 0.8645 | 0.4503 | 0.9267 |

| | | | | | |
|---|---|---|---|---|---|
| RH | 0.80 | 0.90 | 0.8650 | 0.4450 | 0.9283 |

Note: The statistics represent the mean values of R², R, NSD, nuRMSE, and TSS across all combinations in which the respective variable was involved. Bold numbers indicate the highest values, while underlined numbers represent the lowest values.

**Table 5**

Mean evaluation metrics for different combinations of fPAR and water stress indicators at each site.

| Station Name | $R^2$ | R | NSD | nuRMSE | TSS |
|---|---|---|---|---|---|
| HZF | 0.85 | 0.93 | 0.9839 | 0.3685 | 0.9650 |
| MEF | 0.91 | 0.96 | 0.8989 | 0.2918 | 0.9679 |
| CBF | **0.92** | 0.97 | 0.8687 | 0.2716 | 0.9644 |
| QYF | 0.75 | 0.89 | 0.8171 | 0.4677 | 0.9057 |
| ALF | 0.64 | 0.83 | 0.6250 | 0.5950 | 0.7387 |
| DHF | 0.55 | 0.75 | 0.7831 | 0.6671 | 0.8224 |
| BNF | 0.37 | 0.67 | 0.8119 | 0.7540 | 0.8003 |
| YJF | 0.43 | 0.68 | 0.6702 | 0.7348 | 0.7130 |
| XLG | 0.49 | 0.75 | 0.9877 | 0.6980 | 0.8736 |
| NMG | 0.40 | 0.64 | 0.6334 | 0.7685 | 0.6673 |
| DLG | 0.78 | 0.89 | 0.9509 | 0.4543 | 0.9425 |
| DMG | 0.59 | 0.78 | 0.6941 | 0.6204 | 0.7742 |
| HBG_G01 | **0.92** | 0.96 | 0.9040 | 0.2750 | **0.9715** |
| HBG_S01 | 0.53 | 0.82 | 1.1390 | 0.6556 | 0.8945 |
| DXG | 0.17 | 0.83 | 1.3421 | 0.7631 | 0.8326 |
| JZA | 0.80 | 0.91 | 0.7697 | 0.4373 | 0.8916 |
| GCA | 0.62 | 0.82 | 0.9519 | 0.5950 | 0.9014 |
| SYA | 0.81 | 0.92 | 0.7606 | 0.4294 | 0.8854 |
| LCA | 0.64 | 0.80 | 0.7830 | 0.5898 | 0.8488 |
| YCA | 0.64 | 0.80 | 0.7117 | 0.5991 | 0.8043 |
| All | 0.80 | 0.90 | 0.8658 | 0.4450 | 0.9285 |

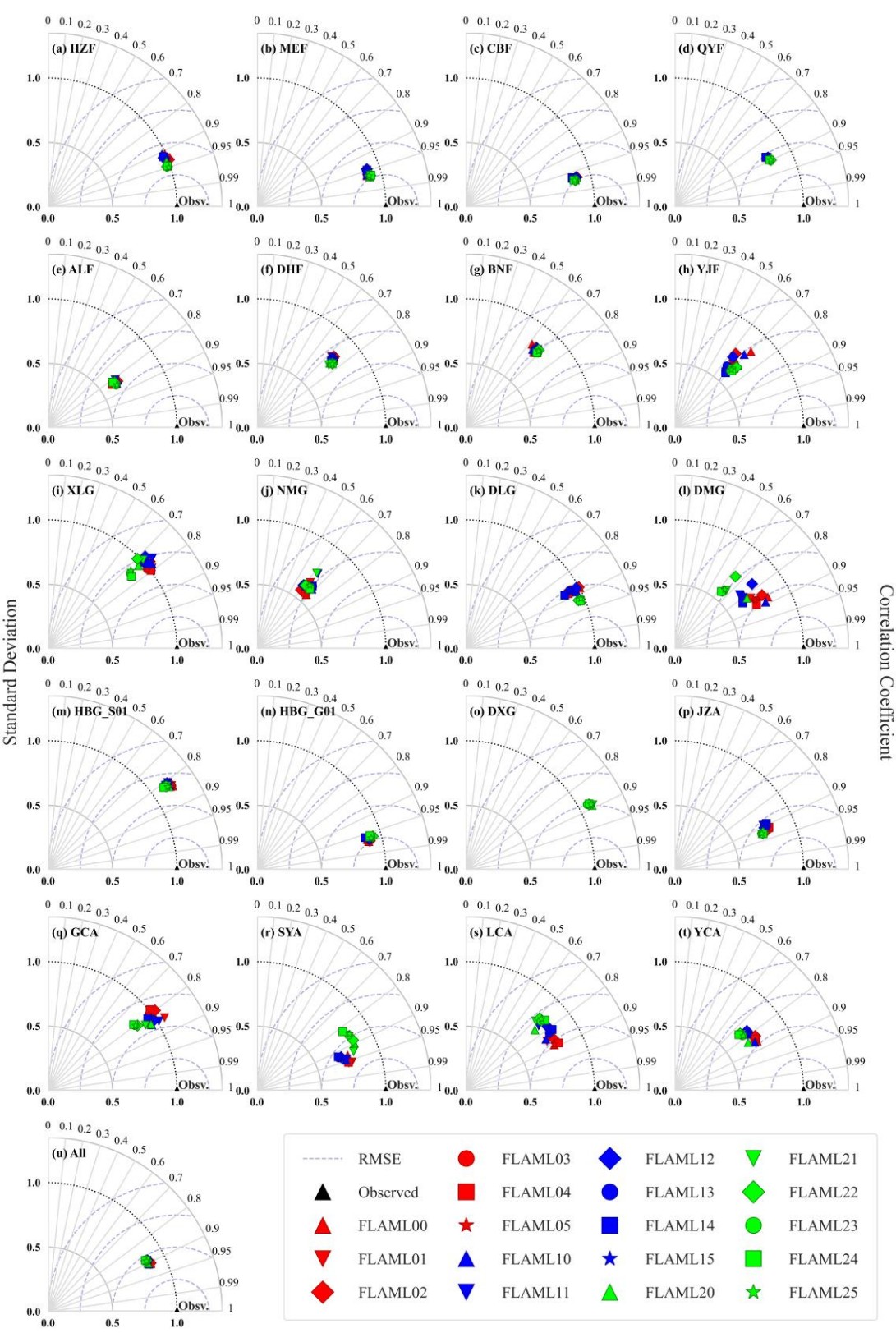


**Figure 3.** Normalized Taylor diagrams showing the performance of the FLAML-LUE model at
various sites based on observed GPP data. Each point represents a specific combination of fPAR
and water stress factor used in the model simulation. Different colors denote different fPAR products:
red for EVI, blue for NDVI, and green for LAI. Marker shapes indicate the type of water stress

factor: "+" for LSWI, "×" for EF, diamond for SW, circle for VPD, square for Pre, and star for RH. Points closer to the reference point (R = 1, NSD = 1) indicate better agreement between simulated and observed GPP. Panels (a)–(h) correspond to eight forest sites, (i)–(o) to seven grassland sites, (p)–(t) to five cropland sites, and (u) presents an overall model evaluation on the validation dataset across all sites.

From an ecosystem perspective, **Table 7** indicate that the FLAML-LUE model achieves the highest fitting accuracy in forest ecosystems ($R^2$ = 0.83, nuRMSE = 0.4162), followed by cropland ecosystems ($R^2$ = 0.72, nuRMSE = 0.5258), and the lowest in grassland ecosystems ($R^2$ = 0.71, nuRMSE = 0.5407). The slope of the fitted line in **Figure 7** is less than 1 for all ecosystem types, indicating that the FLAML-LUE model tends to underestimate GPP, particularly in croplands and grasslands.

**Tables S2, S3,** and **Table 6** collectively demonstrate that the model's performance varies across ecosystem types depending on the choice of fPAR-related variables. In forest ecosystems, the model is relatively insensitive to different fPAR and water-related inputs, with the LAI-driven model achieving the best performance. This can be attributed to LAI's ability to capture forest canopy structure, thereby improving fPAR estimates. In contrast, the model's performance is more sensitive to the choice of input variables in cropland and grassland ecosystems. In croplands, the EVI-driven model performs best, followed by LAI and then NDVI, although the performance differences are moderate. In grasslands, however, the NDVI-driven model performs worst, especially at the DXG site, likely due to NDVI's sensitivity to soil background and saturation in sparse and heterogeneous vegetation. EVI, with reduced saturation and higher sensitivity to biomass, shows better performance in structured cropland areas. Overall, the EVI and LSWI driven model (FLAML00) exhibits the best performance across all ecosystem types.

To further investigate model accuracy across different land cover types, **Figure 5**
presents the R² values of five forest types, three grassland types, and two cropland types
under different models. In general, model performance varies little within the same land
cover type but differs substantially across types. Specifically, DBF, NF, MF, and SC
exhibit higher simulation accuracy, followed by GRA, SHR, and DC, while EBF, SAV,
and MEA perform the worst. These results are consistent with the Taylor diagram in
**Figure 3**.

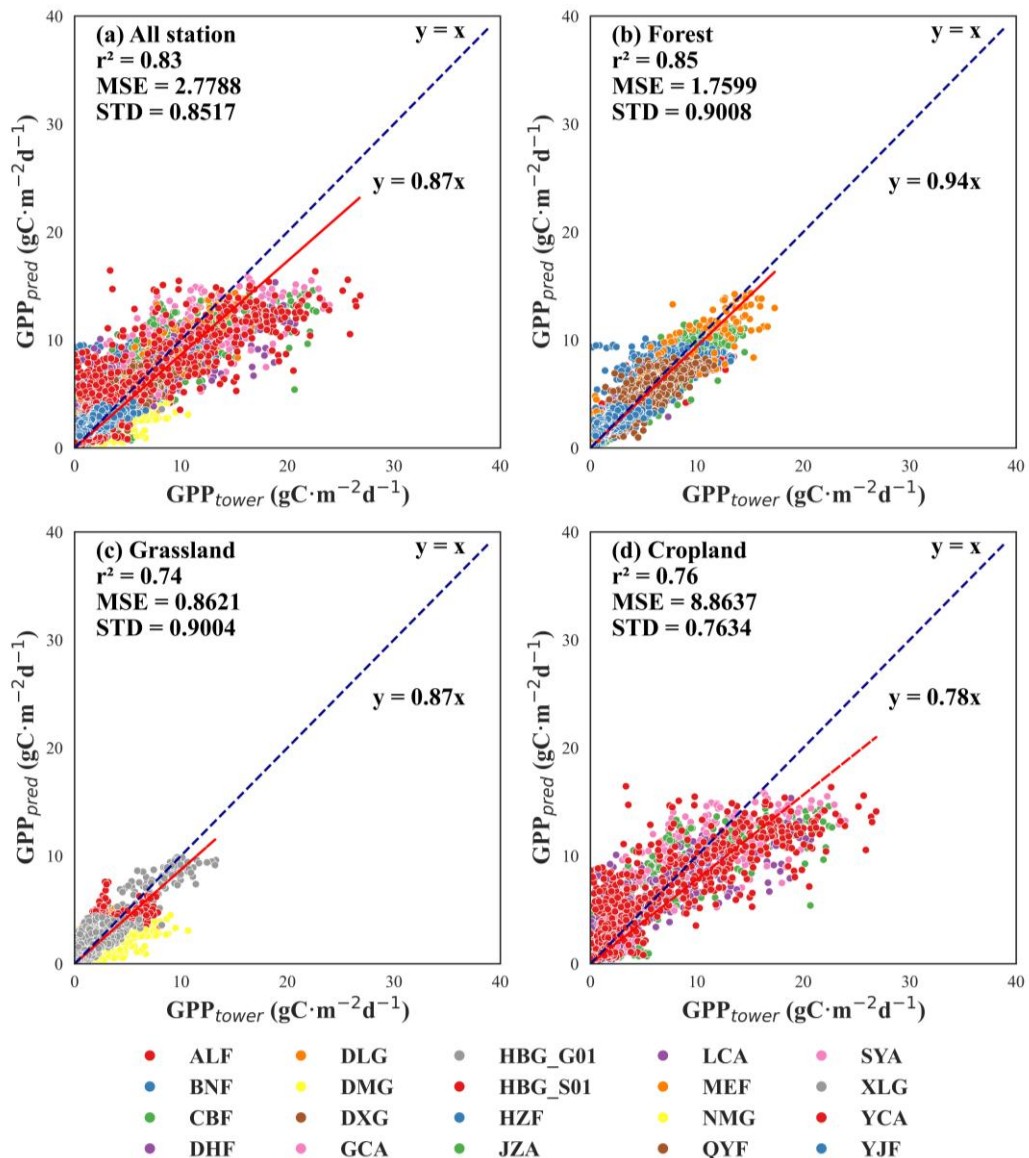


**Figure 4.** Scatterplot of observed GPP vs. simulated GPP. Different colored dots represent different
sites. Note: The simulated GPP values represent the mean of FLAML00 to FLAML25.


**Table 6**

Summary of evaluation metrics for FLAML-LUE model performance across all validation sites.

| FLAML | $R^2$ | | | TSS | | |
|---|---|---|---|---|---|---|
| | Forest | Grass | Crop | Forest | Grass | Crop |
| FLAML00 | 0.83 | 0.73 | 0.77 | 0.9476 | 0.9241 | 0.9004 |
| FLAML01 | 0.84 | 0.71 | 0.75 | 0.9472 | 0.9187 | 0.8946 |
| FLAML02 | 0.84 | 0.70 | 0.75 | 0.9516 | 0.9167 | 0.8966 |
| FLAML03 | 0.84 | 0.71 | 0.76 | 0.9485 | 0.9169 | 0.8971 |
| FLAML04 | 0.84 | 0.72 | 0.76 | 0.9475 | 0.9157 | 0.8991 |
| FLAML05 | 0.84 | 0.72 | 0.76 | 0.9487 | 0.9171 | 0.8927 |
| FLAML10 | 0.83 | 0.72 | 0.76 | 0.9463 | 0.9213 | 0.8861 |
| FLAML11 | 0.83 | 0.68 | 0.70 | 0.9464 | 0.9124 | 0.8696 |
| FLAML12 | 0.84 | 0.67 | 0.70 | 0.9487 | 0.9091 | 0.8717 |
| FLAML13 | 0.83 | 0.69 | 0.71 | 0.9459 | 0.9083 | 0.8696 |
| FLAML14 | 0.83 | 0.69 | 0.70 | 0.9450 | 0.9060 | 0.8713 |
| FLAML15 | 0.84 | 0.69 | 0.71 | 0.9486 | 0.9096 | 0.8746 |
| FLAML20 | 0.85 | 0.73 | 0.73 | 0.9525 | 0.9219 | 0.8718 |
| FLAML21 | 0.85 | 0.71 | 0.70 | 0.9531 | 0.9186 | 0.8575 |
| FLAML22 | 0.86 | 0.70 | 0.68 | 0.9549 | 0.9150 | 0.8545 |
| FLAML23 | 0.86 | 0.71 | 0.69 | 0.9539 | 0.9153 | 0.8535 |
| FLAML24 | 0.85 | 0.72 | 0.67 | 0.9532 | 0.9145 | 0.8465 |
| FLAML25 | 0.86 | 0.71 | 0.68 | 0.9542 | 0.9163 | 0.8561 |
| Statistics | | | | | | |
| EVI | 0.84 | **0.72** | **0.76** | 0.9485 | 0.9182 | 0.8968 |
| NDVI | 0.83 | 0.69 | 0.72 | 0.9468 | 0.9111 | 0.8738 |
| LAI | **0.85** | 0.71 | 0.69 | 0.9536 | 0.9169 | 0.8566 |
| LSWI | 0.84 | **0.73** | **0.75** | 0.9488 | 0.9224 | 0.8861 |
| EF | 0.84 | 0.70 | 0.72 | 0.9489 | 0.9166 | 0.8739 |
| SW | 0.84 | 0.69 | 0.71 | 0.9517 | 0.9136 | 0.8743 |
| VPD | 0.84 | 0.70 | 0.72 | 0.9495 | 0.9135 | 0.8734 |
| Pre | 0.84 | 0.71 | 0.71 | 0.9486 | 0.9121 | 0.8723 |
| RH | 0.84 | 0.70 | 0.72 | 0.9505 | 0.9143 | 0.8745 |


**Table 7**
Mean evaluation metrics for different combinations of fPAR and water stress indicators across
various ecosystems.

| Ecosystem | $R^2$ | R | $\hat{\sigma}_f$ | nuRMSE | TSS |
|-----------|-------|-----|------------------|--------|------|
| ALL | 0.80 | 0.90 | 0.8658 | 0.4450 | 0.9285 |
| Forest | **0.83** | 0.91 | 0.8958 | 0.4162 | **0.9431** |
| Grassland | 0.71 | 0.84 | 0.9187 | 0.5407 | 0.9154 |
| Croplands | 0.72 | 0.85 | 0.7893 | 0.5258 | 0.8757 |

Note: The evaluation metrics for all sites and different ecosystem types were calculated based on
the average of 18 simulation results.
Regarding $CV_{RMSE}$, SHR shows the largest error, followed by MEA, GRA, SC, and
DC, while the five forest types show the smallest errors. This may be attributed to the
greater GPP variability in grassland and cropland ecosystems, which are more strongly
influenced by climatic variability and anthropogenic activities, leading to higher model
uncertainty. In contrast, forest ecosystems have more stable structures and continuous
carbon exchange processes, resulting in more robust model performance. Although
alpine meadow is classified as grassland ecosystems, their extreme climatic conditions,
short growing season, and high sensitivity to temperature and precipitation further
increase the uncertainty of GPP simulation, leading to higher errors.
In terms of PBias, SHR consistently shows a pronounced overestimation across all
models. Similarly, SAV and MEA are also generally overestimated in all models,
though to a lesser extent than SHR. EBF exhibits a slight overestimation as well. Other
vegetation types display only minor underestimation or overestimation. Overall, the
models perform best for DBF, NF, and MF, followed by EBF, MEA, SC, and DC, while
the simulation accuracy is relatively poor for SAV, SC, and especially SHR.
Biases also differ among grassland ecosystems, especially for typical grasslands,
alpine meadows, and shrublands. Typical grasslands tend to be underestimated, while
alpine meadows and shrublands are often overestimated. These biases may result from
the model's limited ability to capture seasonal changes in water availability and its
interaction with temperature. Typical grasslands usually show high productivity when
water is sufficient, especially in spring and summer. If the model fails to reflect these
seasonal patterns, it can lead to underestimation. In contrast, productivity in alpine
meadows is mainly limited by low temperatures and a short growing season. If the
model does not fully consider these constraints, it may overestimate photosynthesis and
thus GPP. For shrublands, overestimation may be due to high spatial heterogeneity,
including a mix of shrubs, grasses, and bare soil. This complexity is difficult to capture
in remote sensing data (e.g., fPAR) and model inputs, leading to possible overestimation
of productivity.

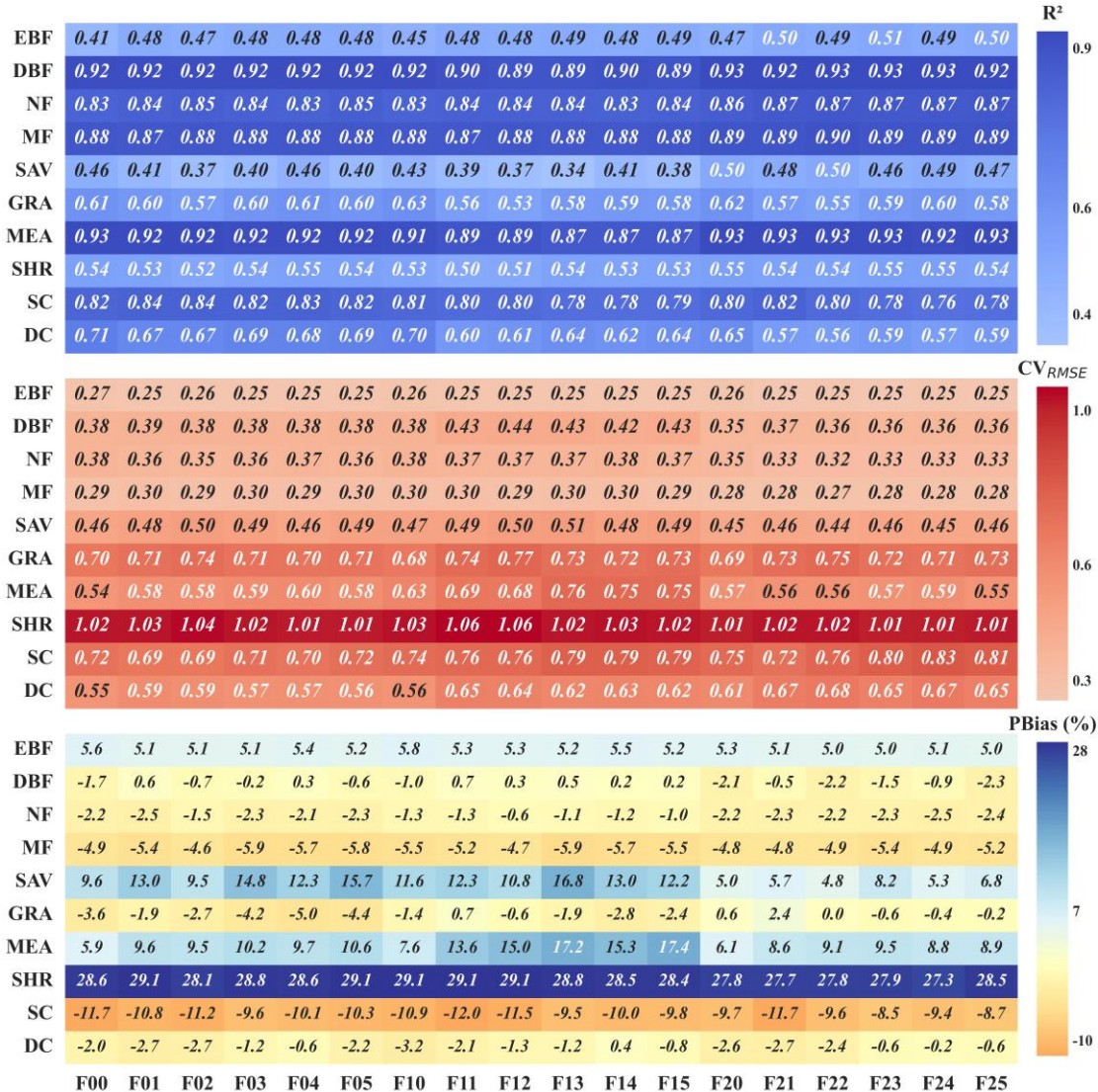

| | F00 | F01 | F02 | F03 | F04 | F05 | F10 | F11 | F12 | F13 | F14 | F15 | F20 | F21 | F22 | F23 | F24 | F25 |
|---|---|---|---|---|---|---|---|---|---|---|---|---|---|---|---|---|---|---|
| **R²** | | | | | | | | | | | | | | | | | | |
| EBF | 0.41 | 0.48 | 0.47 | 0.48 | 0.48 | 0.48 | 0.45 | 0.48 | 0.48 | 0.49 | 0.48 | 0.49 | 0.47 | 0.50 | 0.49 | 0.51 | 0.49 | 0.50 |
| DBF | 0.92 | 0.92 | 0.92 | 0.92 | 0.92 | 0.92 | 0.92 | 0.90 | 0.89 | 0.89 | 0.90 | 0.89 | 0.93 | 0.92 | 0.93 | 0.93 | 0.93 | 0.92 |
| NF | 0.83 | 0.84 | 0.85 | 0.84 | 0.83 | 0.85 | 0.83 | 0.84 | 0.84 | 0.84 | 0.83 | 0.84 | 0.86 | 0.87 | 0.87 | 0.87 | 0.87 | 0.87 |
| MF | 0.88 | 0.87 | 0.88 | 0.88 | 0.88 | 0.88 | 0.88 | 0.87 | 0.88 | 0.88 | 0.88 | 0.88 | 0.89 | 0.89 | 0.90 | 0.89 | 0.89 | 0.89 |
| SAV | 0.46 | 0.41 | 0.37 | 0.40 | 0.46 | 0.40 | 0.43 | 0.39 | 0.37 | 0.34 | 0.41 | 0.38 | 0.50 | 0.48 | 0.50 | 0.46 | 0.49 | 0.47 |
| GRA | 0.61 | 0.60 | 0.57 | 0.60 | 0.61 | 0.60 | 0.63 | 0.56 | 0.53 | 0.58 | 0.59 | 0.58 | 0.62 | 0.57 | 0.55 | 0.59 | 0.60 | 0.58 |
| MEA | 0.93 | 0.92 | 0.92 | 0.92 | 0.92 | 0.92 | 0.91 | 0.89 | 0.89 | 0.87 | 0.87 | 0.87 | 0.93 | 0.93 | 0.93 | 0.93 | 0.92 | 0.93 |
| SHR | 0.54 | 0.53 | 0.52 | 0.54 | 0.55 | 0.54 | 0.53 | 0.50 | 0.51 | 0.54 | 0.53 | 0.53 | 0.55 | 0.54 | 0.54 | 0.55 | 0.55 | 0.54 |
| SC | 0.82 | 0.84 | 0.84 | 0.82 | 0.83 | 0.82 | 0.81 | 0.80 | 0.80 | 0.78 | 0.78 | 0.79 | 0.80 | 0.82 | 0.80 | 0.78 | 0.76 | 0.78 |
| DC | 0.71 | 0.67 | 0.67 | 0.69 | 0.68 | 0.69 | 0.70 | 0.60 | 0.61 | 0.64 | 0.62 | 0.64 | 0.65 | 0.57 | 0.56 | 0.59 | 0.57 | 0.59 |
| **CV$_{RMSE}$** | | | | | | | | | | | | | | | | | | |
| EBF | 0.27 | 0.25 | 0.26 | 0.25 | 0.25 | 0.25 | 0.26 | 0.25 | 0.25 | 0.25 | 0.25 | 0.25 | 0.26 | 0.25 | 0.25 | 0.25 | 0.25 | 0.25 |
| DBF | 0.38 | 0.39 | 0.38 | 0.38 | 0.38 | 0.38 | 0.38 | 0.43 | 0.44 | 0.43 | 0.42 | 0.43 | 0.35 | 0.37 | 0.36 | 0.36 | 0.36 | 0.36 |
| NF | 0.38 | 0.36 | 0.35 | 0.36 | 0.37 | 0.36 | 0.38 | 0.37 | 0.37 | 0.37 | 0.38 | 0.37 | 0.35 | 0.33 | 0.32 | 0.33 | 0.33 | 0.33 |
| MF | 0.29 | 0.30 | 0.29 | 0.30 | 0.29 | 0.30 | 0.30 | 0.30 | 0.29 | 0.30 | 0.30 | 0.29 | 0.28 | 0.28 | 0.27 | 0.28 | 0.28 | 0.28 |
| SAV | 0.46 | 0.48 | 0.50 | 0.49 | 0.46 | 0.49 | 0.47 | 0.49 | 0.50 | 0.51 | 0.48 | 0.49 | 0.45 | 0.46 | 0.44 | 0.46 | 0.45 | 0.46 |
| GRA | 0.70 | 0.71 | 0.74 | 0.71 | 0.70 | 0.71 | 0.68 | 0.74 | 0.77 | 0.73 | 0.72 | 0.73 | 0.69 | 0.73 | 0.75 | 0.72 | 0.71 | 0.73 |
| MEA | 0.54 | 0.58 | 0.58 | 0.59 | 0.60 | 0.58 | 0.63 | 0.69 | 0.68 | 0.76 | 0.75 | 0.75 | 0.57 | 0.56 | 0.56 | 0.57 | 0.59 | 0.55 |
| SHR | 1.02 | 1.03 | 1.04 | 1.02 | 1.01 | 1.01 | 1.03 | 1.06 | 1.06 | 1.02 | 1.03 | 1.02 | 1.01 | 1.02 | 1.02 | 1.01 | 1.01 | 1.01 |
| SC | 0.72 | 0.69 | 0.69 | 0.71 | 0.70 | 0.72 | 0.74 | 0.76 | 0.76 | 0.79 | 0.79 | 0.79 | 0.75 | 0.72 | 0.76 | 0.80 | 0.83 | 0.81 |
| DC | 0.55 | 0.59 | 0.59 | 0.57 | 0.57 | 0.56 | 0.56 | 0.65 | 0.64 | 0.62 | 0.63 | 0.62 | 0.61 | 0.67 | 0.68 | 0.65 | 0.67 | 0.65 |
| **PBias (%)** | | | | | | | | | | | | | | | | | | |
| EBF | 5.6 | 5.1 | 5.1 | 5.1 | 5.4 | 5.2 | 5.8 | 5.3 | 5.3 | 5.2 | 5.5 | 5.2 | 5.3 | 5.1 | 5.0 | 5.0 | 5.1 | 5.0 |
| DBF | -1.7 | 0.6 | -0.7 | -0.2 | 0.3 | -0.6 | -1.0 | 0.7 | 0.3 | 0.5 | 0.2 | 0.2 | -2.1 | -0.5 | -2.2 | -1.5 | -0.9 | -2.3 |
| NF | -2.2 | -2.5 | -1.5 | -2.3 | -2.1 | -2.3 | -1.3 | -1.3 | -0.6 | -1.1 | -1.2 | -1.0 | -2.2 | -2.3 | -2.2 | -2.3 | -2.5 | -2.4 |
| MF | -4.9 | -5.4 | -4.6 | -5.9 | -5.7 | -5.8 | -5.5 | -5.2 | -4.7 | -5.9 | -5.7 | -5.5 | -4.8 | -4.8 | -4.9 | -5.4 | -4.9 | -5.2 |
| SAV | 9.6 | 13.0 | 9.5 | 14.8 | 12.3 | 15.7 | 11.6 | 12.3 | 10.8 | 16.8 | 13.0 | 12.2 | 5.0 | 5.7 | 4.8 | 8.2 | 5.3 | 6.8 |
| GRA | -3.6 | -1.9 | -2.7 | -4.2 | -5.0 | -4.4 | -1.4 | 0.7 | -0.6 | -1.9 | -2.8 | -2.4 | 0.6 | 2.4 | 0.0 | -0.6 | -0.4 | -0.2 |
| MEA | 5.9 | 9.6 | 9.5 | 10.2 | 9.7 | 10.6 | 7.6 | 13.6 | 15.0 | 17.2 | 15.3 | 17.4 | 6.1 | 8.6 | 9.1 | 9.5 | 8.8 | 8.9 |
| SHR | 28.6 | 29.1 | 28.1 | 28.8 | 28.6 | 29.1 | 29.1 | 29.1 | 29.1 | 28.8 | 28.5 | 28.4 | 27.8 | 27.7 | 27.8 | 27.9 | 27.3 | 28.5 |
| SC | -11.7 | -10.8 | -11.2 | -9.6 | -10.1 | -10.3 | -10.9 | -12.0 | -11.5 | -9.5 | -10.0 | -9.8 | -9.7 | -11.7 | -9.6 | -8.5 | -9.4 | -8.7 |
| DC | -2.0 | -2.7 | -2.7 | -1.2 | -0.6 | -2.2 | -3.2 | -2.1 | -1.3 | -1.2 | 0.4 | -0.8 | -2.6 | -2.7 | -2.4 | -0.6 | -0.2 | -0.6 |

**Figure 5.** Comparison of R², CV$_{RMSE}$, and PBias of GPP estimates from different FLAML-LUE models across various land cover types. Note: F00 represents FLAML00, and so on.

Across the four temporal scales, the performance of the 18 FLAML-LUE models improves as the temporal resolution becomes coarser. The average R² across 20 sites increases from 0.64 at the daily scale to 0.74 at the monthly scale (**Table S8**), while the average nuRMSE decreases from 0.5518 to 0.4088. Paired t-tests show that, except for YJF, NMG, DMG, DXG, and YCA, the FLAML-LUE model exhibits significantly lower R² at the daily scale than at longer temporal scales ($p < 0.05$, **Figure 6**). For these five sites, model performance remains relatively stable across different temporal scales.

Furthermore, compared to the daily scale, the nuRMSE decreases by 12.97%,

16.52%, and 25.92% at the 8-day, 16-day, and monthly scales, respectively, indicating
that the uncertainty of the FLAML-LUE model is significantly reduced at coarser
temporal resolutions.

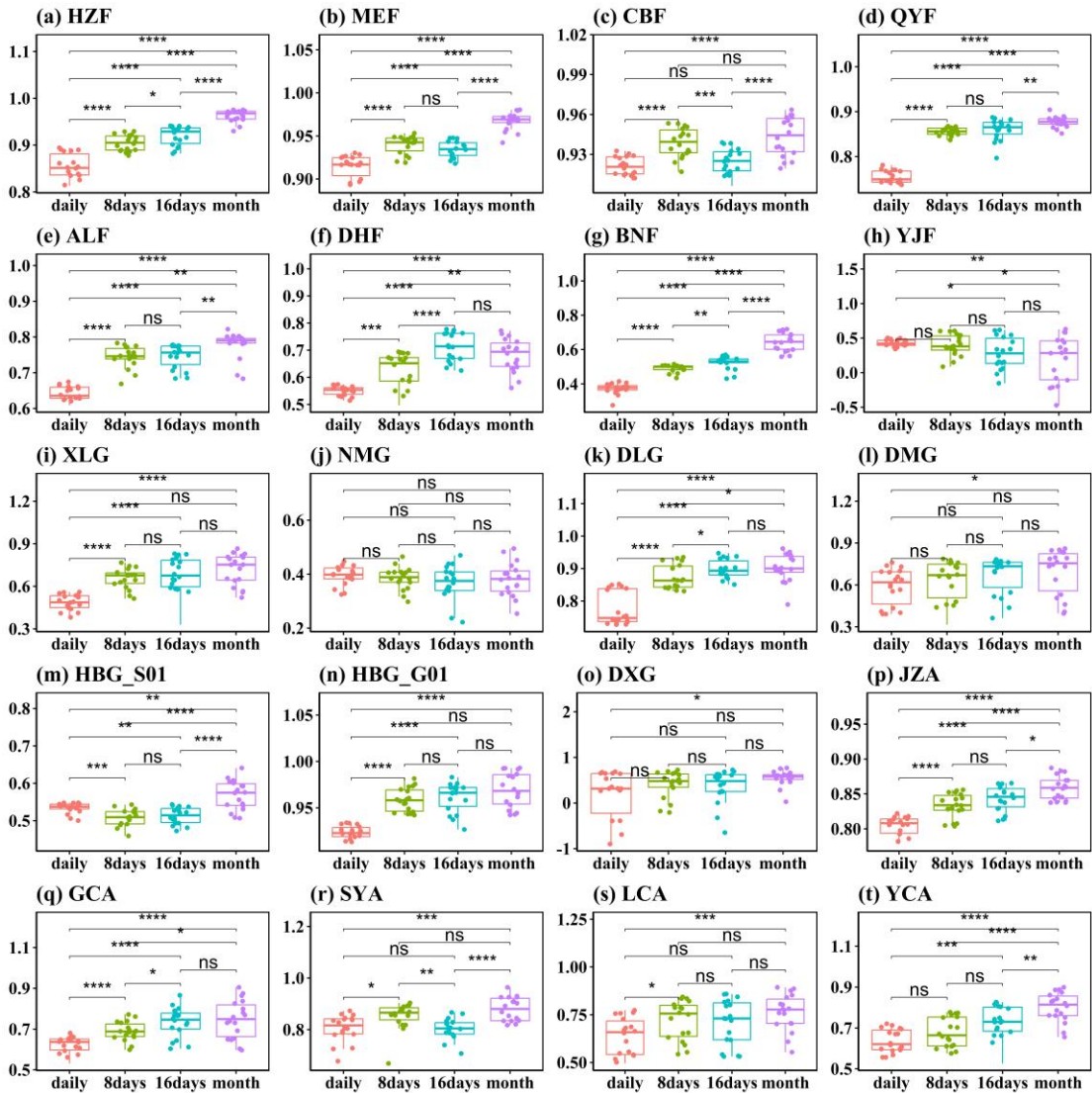


**Figure 6**. Asterisks indicate significant differences between the $R^2$ at the four temporal resolutions (Kruskal-Wallis test), ****p values < 0.0001, ***p values < 0.001, **p values < 0.01, *p values ⩽ 0.05, and ns indicates no significance (p > 0.05).

Overall, the accuracy of FLAML-LUE models constructed using different
combinations of fPAR and water stress indicators showed limited variation, with the
FLAML00 model (fPAR = EVI, water = LSWI) demonstrating the best performance.
However, the model exhibited considerable differences in performance across

ecosystem types, with the highest accuracy observed in forest ecosystems, followed by

croplands and then grasslands. Further analysis by specific vegetation cover types

revealed that the model performed best for DBF, NF, and MF, followed by GRA, MEA

SC, and DC, while its performance was relatively poor for EBF, SAV, and particularly

SHR (PBias > 27%, CVrmse > 1, $R^2$ < 0.6). In addition, evaluation across different

temporal scales indicated that model uncertainty decreased with increasing time

intervals, suggesting that the FLAML-LUE model exhibits greater robustness and

reliability at coarser temporal resolutions.

## 3.2 Model Evaluation Under Extreme Climatic Conditions

Numerous studies have demonstrated that climate extremes such as heatwaves,

droughts, and high atmospheric VPD can substantially alter ecosystem dynamics and

reduce carbon uptake capacity (Frank et al., 2015; Reichstein et al., 2013). These

extreme events can suppress photosynthesis, increase respiration, and disrupt the

balance of carbon exchange between vegetation and the atmosphere. In order to

evaluate the robustness and reliability of the FLAML-LUE models under such stress

conditions, this study further investigates model performance in simulating GPP under

three types of climate extremes: high temperature, high VPD, and drought. By

analyzing the response of model accuracy and bias under these scenarios, we aim to

assess its applicability and limitations in extreme environmental conditions.

## 3.2.1 Performance Under High Temperature Events

**Figure 7** shows the performance of 18 FLAML-LUE models under high-

temperature and non-high-temperature conditions. The results indicate a significant

decline in model accuracy under high-temperature conditions. As shown in **Figure 7a**, the models perform well under non-high-temperature conditions, with the R values of all 18 FLAML-LUE models exceeding 0.9. However, under high-temperature conditions, the Taylor diagram reveals a significant decrease in model performance, with correlation coefficients dropping and a substantial increase in nuRMSE, indicating a reduced ability to capture GPP dynamics.

Interestingly, as shown in **Figure 7b**, the $CV_{RMSE}$ values under non-high-temperature conditions are generally higher than under high-temperature conditions. This may be due to higher observed GPP values under high temperatures, resulting in a larger denominator for $CV_{RMSE}$, which can reduce the $CV_{RMSE}$ despite larger prediction errors. Overall, the difference in prediction bias between high-temperature and non-high-temperature conditions is minimal.

**Figure 7c** shows that, under high-temperature conditions, the PBias fluctuates more significantly, with more stations showing severe overestimation or underestimation. Specifically, some models (e.g., FLAML00, FLAML01, FLAML11, FLAML15, FLAML21) overestimate GPP at certain sites under high-temperature conditions, while all models show more severe underestimation at other sites. Models driven by LAI (FLAML20 – FLAML25) exhibit smaller bias variations under non-high-temperature conditions, with PBias mainly ranging from -0.3 to 0.3.

In conclusion, high-temperature conditions increase model uncertainty, with all models exhibiting varying degrees of overestimation or underestimation across sites. Models incorporating VPD, precipitation, and relative humidity as water stress factors

perform better overall, indicating greater robustness under high-temperature stress.

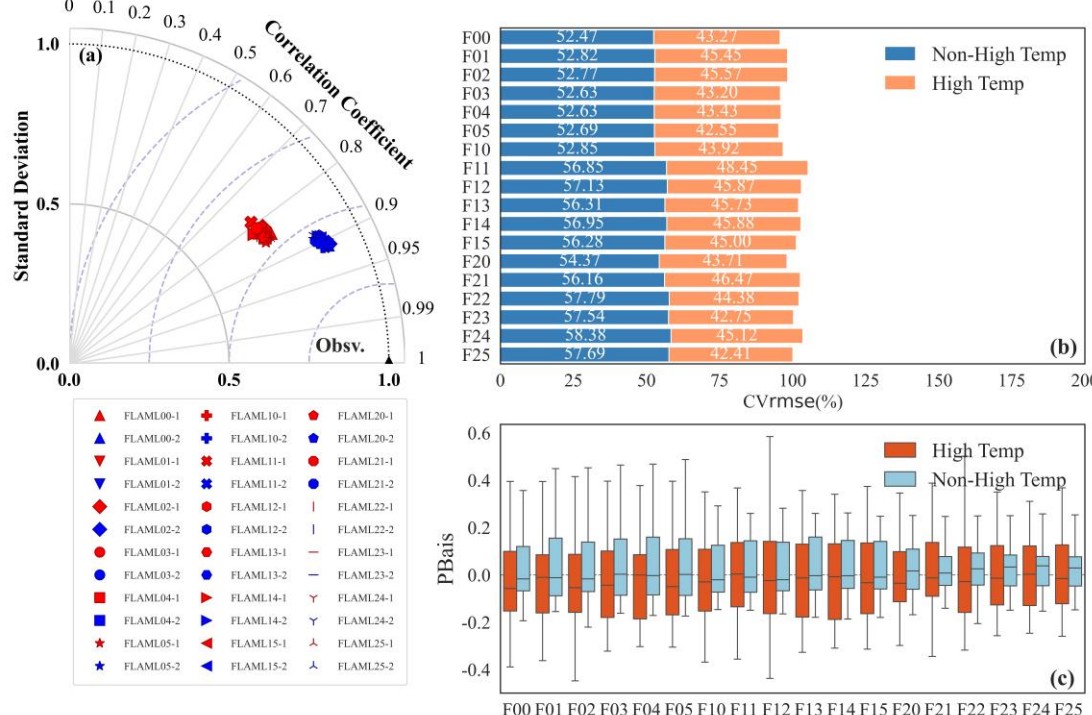


**Figure 7**. The comparison of GPP products performance under high temperature and non-high
temperature (In the Taylor diagram, 1 represents high temperature, 2 represents non-high
temperature).
Differences in model performance under high-temperature and non-high-
temperature conditions are pronounced across various land cover types. **Figure 8**
compares the estimation accuracy of different land cover types under both conditions.
Overall, model accuracy in simulating GPP is significantly lower under high-
temperature conditions, with $R^2$ values showing a notable decline. Specifically, for the
NF type, the $R^2$ under high temperatures approaches a negative value, indicating very
low explanatory power, whereas under non-high-temperature conditions, $R^2$ ranges
from 0.83 to 0.87. Notably, the FLAML13 model for Savannas shows a drastic decrease
in $R^2$ from 0.38 under non-high-temperature conditions to -1.46 under high-temperature
conditions, performing even worse than the mean of the data during high temperatures.
Corresponding to **Figure 7**, $CV_{RMSE}$ is generally lower under high-temperature
conditions than under non-high-temperature conditions. The SHR type exhibits a higher
coefficient of variation, while PBias shows more pronounced fluctuations. For SHR
and EBF, the models tend to overestimate GPP under both temperature conditions, with
overestimation more pronounced under high temperatures. In contrast, MEA shows
underestimation under high-temperature conditions but overestimation under non-high-
temperature conditions. Overall, most land cover types exhibit a greater degree of
underestimation under high-temperature conditions. Nevertheless, the MF type
maintains relatively high simulation accuracy. In contrast, the NBF, NF, and SC types
are more strongly affected by high temperatures, with NF showing negative simulation
accuracy under high-temperature conditions and SC exhibiting marked variations in
PBias.

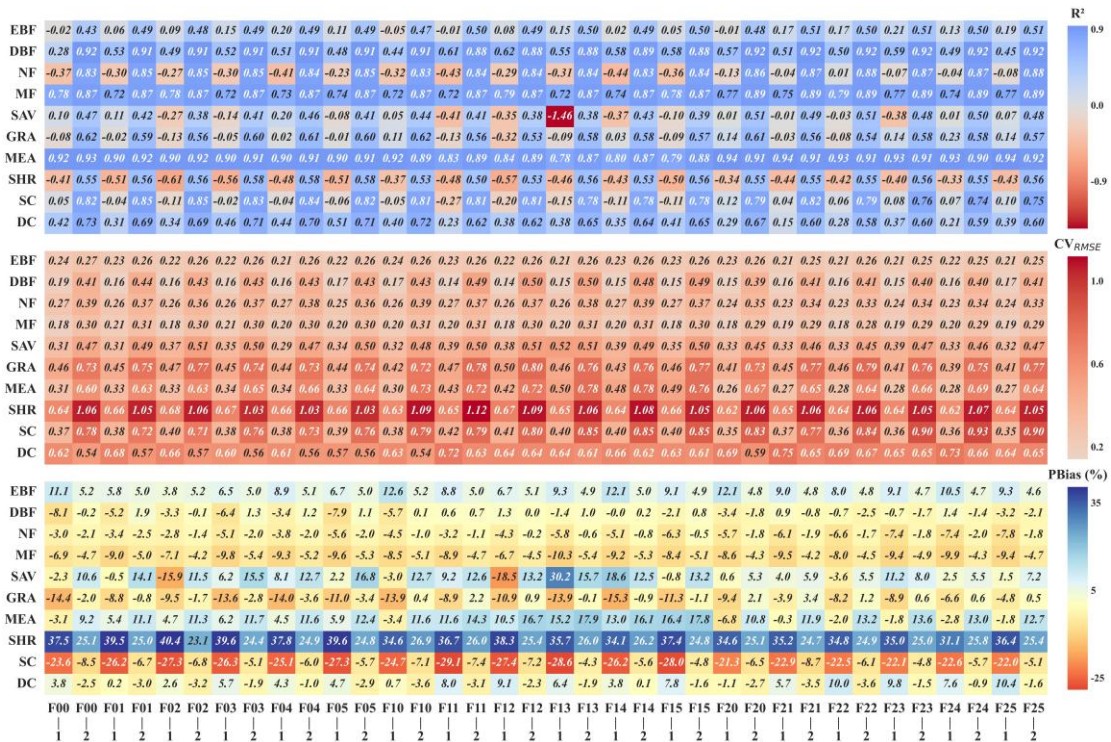


**Figure 8.** Comparison of statistical indicators ($R^2$, $CV_{RMSE}$, PBias) of FLAML-LUE model under high temperature conditions and non-high temperature conditions for different land cover types (1 represents high temperature, 2 represents non-high temperature).

## 3.2.2 Performance Under High VPD

**Figure 9** shows the performance of the 18 FLAML-LUE models under high and non-high VPD conditions. Unlike the high-temperature scenario, the statistical metrics of all models exhibit only a slight decline under high VPD, indicating a less pronounced impact on model performance. As shown in **Figure 9a**, the variability in model performance increases under high VPD conditions. However, **Figure 9b** reveals that CVRMSE values are generally higher under non-high VPD conditions, a trend consistent with the results observed under high-temperature conditions.

Under high VPD, PBias exhibits significant fluctuations compared to non-high VPD conditions (**Figure 9c**). Specifically, the average PBias across sites is higher under high VPD, whereas it is lower under non-high VPD. In high VPD conditions, models

driven by EVI show smaller differences in PBias across sites, with values primarily

ranging from -0.4 to 0.5. In contrast, FLAML05 shows larger differences in PBias

between sites under non-high VPD, with overestimations at some sites. Overall, model

performance under high VPD shows greater uncertainty, with both overestimations and

underestimations occurring across different sites. In general, EVI-driven models

perform more consistently under both high and non-high VPD conditions.

Model performance also differs across land cover types under high and non-high

VPD conditions. **Figure 10** compares the estimation accuracy for various land cover

types under both conditions. Overall, GPP simulation accuracy for certain cover types

(e.g., DBF, MF, MEA, SC, DC) shows little difference between high and non-high VPD

conditions. Although $R^2$ values for some land cover types are significantly lower under

high VPD than under non-high VPD, the impact of high VPD on model performance is

smaller compared to high temperature. The most notable example is the FLAML13

model for Savannas, where $R^2$ drops significantly from -1.46 under non-high VPD to -

0.39 under high VPD, performing worse than the mean data value under high VPD.

Similar to high-temperature conditions, $CV_{RMSE}$ under high VPD is generally lower

than under non-high VPD. MEA shows a larger coefficient of variation, and PBias

exhibits more noticeable fluctuations. For the EBF and SHR type, models tend to

overestimate GPP in both high and non-high VPD conditions, with the overestimation

being more pronounced under high VPD. SC and GRA models show significant

underestimation under high VPD. DBF, NF, and MF perform relatively well under high

VPD, while SC underestimates GPP under both conditions, and DC overestimates GPP

under high VPD but underestimates it under non-high VPD. Overall, compared to high-
temperature conditions, the effect of high VPD on estimation errors is smaller across
different land cover types.

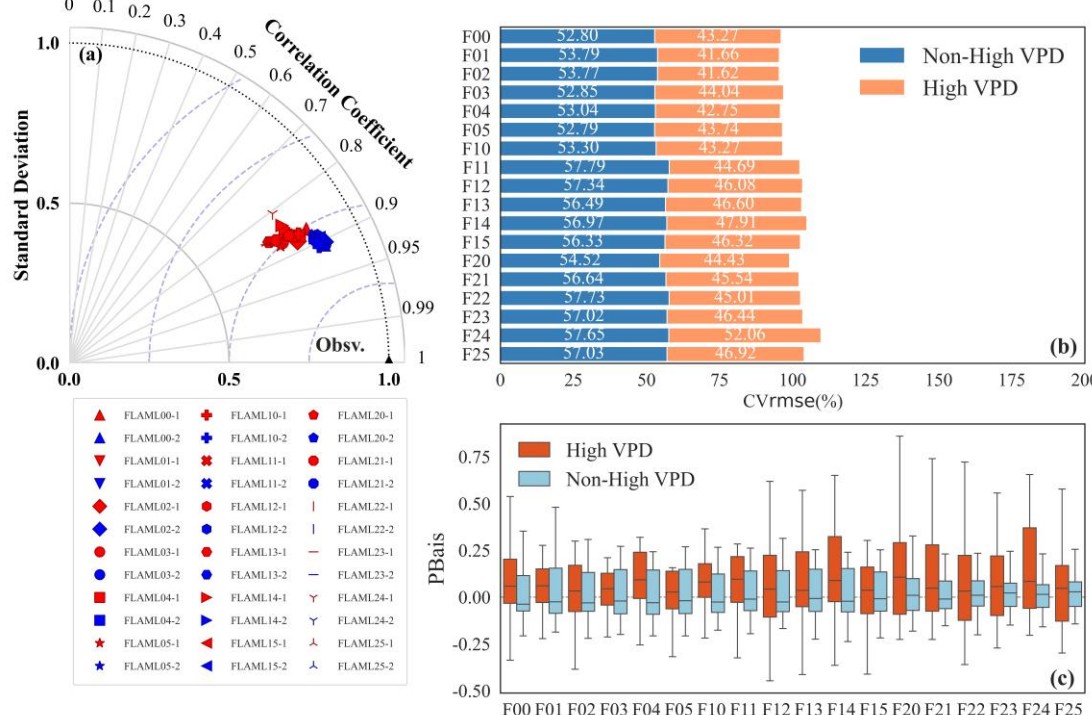


**Figure 9**. The comparison of GPP products performance under high VPD and non-high VPD (In
the Taylor diagram, 1 represents high VPD, 2 represents non-high VPD).

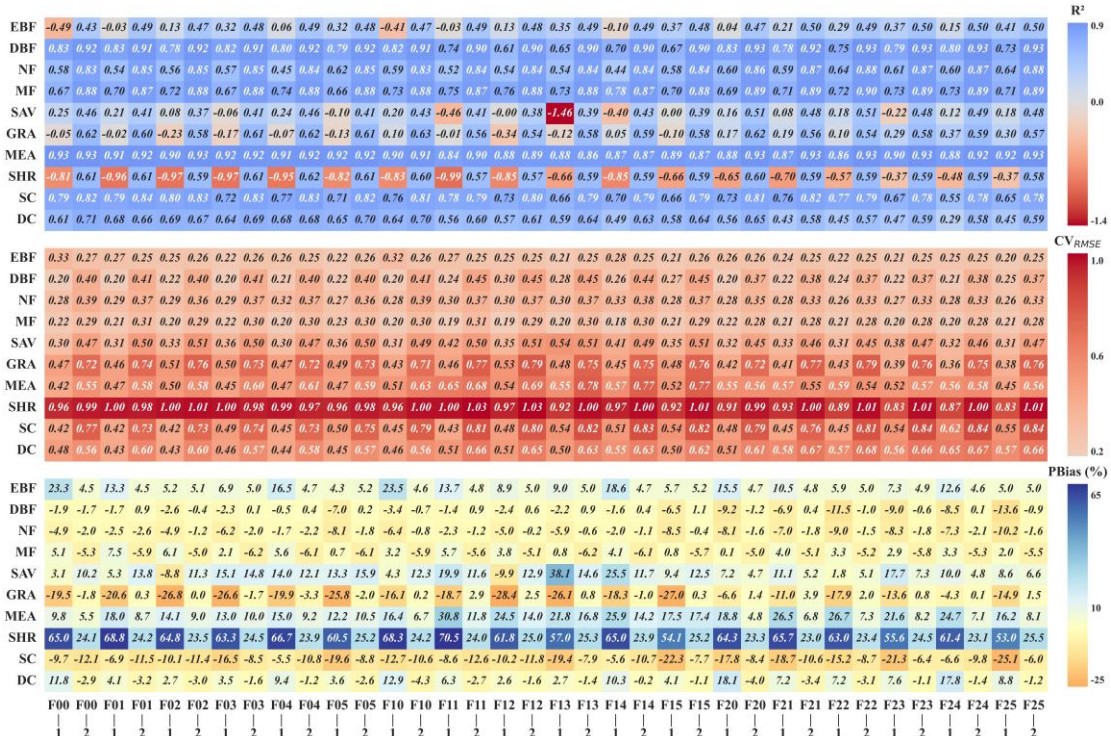


**Figure 10**. Comparison of statistical indicators ($R^2$, $CV_{RMSE}$, PBias) of FLAML-LUE model under high VPD conditions and non-high VPD conditions for different land cover types (1 represents high VPD, 2 represents non-high VPD).

### 3.2.3 Performance Under Drought Conditions

**Figure 11** presents the simulation performance of the 18 FLAML-LUE models under drought and non-drought conditions. Unlike the decline in performance under high temperature and high VPD conditions, the model shows similar or even slightly better accuracy under drought compared to non-drought conditions. This may be attributed to an overall reduction in GPP and its variability during drought periods, which potentially makes it easier for the models to capture the general trend and thereby improves simulation accuracy.

Compared to the boxplots under non-drought conditions, drought notably increases the variability in PBias across sites for all models, particularly due to substantial overestimation at certain sites. In contrast, the degree of underestimation remains

similar to that under non-drought conditions. Among the models, those driven by EVI exhibit the best overall performance, followed by those using LAI as the vegetation indicator.

**Figure 12** shows that drought substantially affects GPP estimation accuracy across most land cover types. For certain types, such as savannas and deciduous broadleaf forests, no data were available during drought months, making performance evaluation under drought impossible. For other land cover types, the impact of drought varies significantly. Specifically, EBF, MEA, and DC show higher $R^2$ values under drought, while NF, MF, GRA, SHR, and SC perform better under non-drought conditions. Among them, MF and SHR have the lowest simulation accuracy under drought but perform relatively well during non-drought periods.

Regarding $CV_{RMSE}$, all land cover types except MEA and NF exhibit lower values under drought conditions, consistent with the results in **Figure 11a**. MEA shows the largest coefficient of variation, indicating greater variability in model performance under drought. In terms of PBias, NF, MEA, and SHR exhibit the highest errors. On average, model errors increase under drought across most land cover types. Except for EBF and GRA, most types show severe overestimation or underestimation during drought periods.

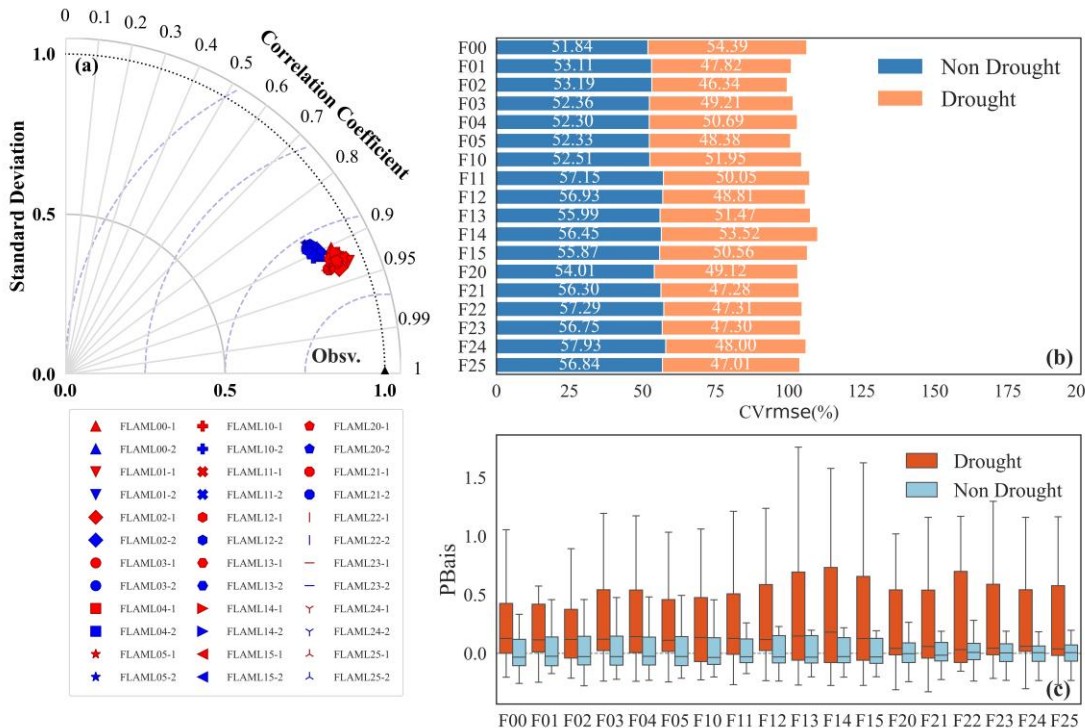

**Figure 11**. The comparison of GPP products performance under drought and non-drought (In the
Taylor diagram, 1 represents drought, 2 represents non drought).

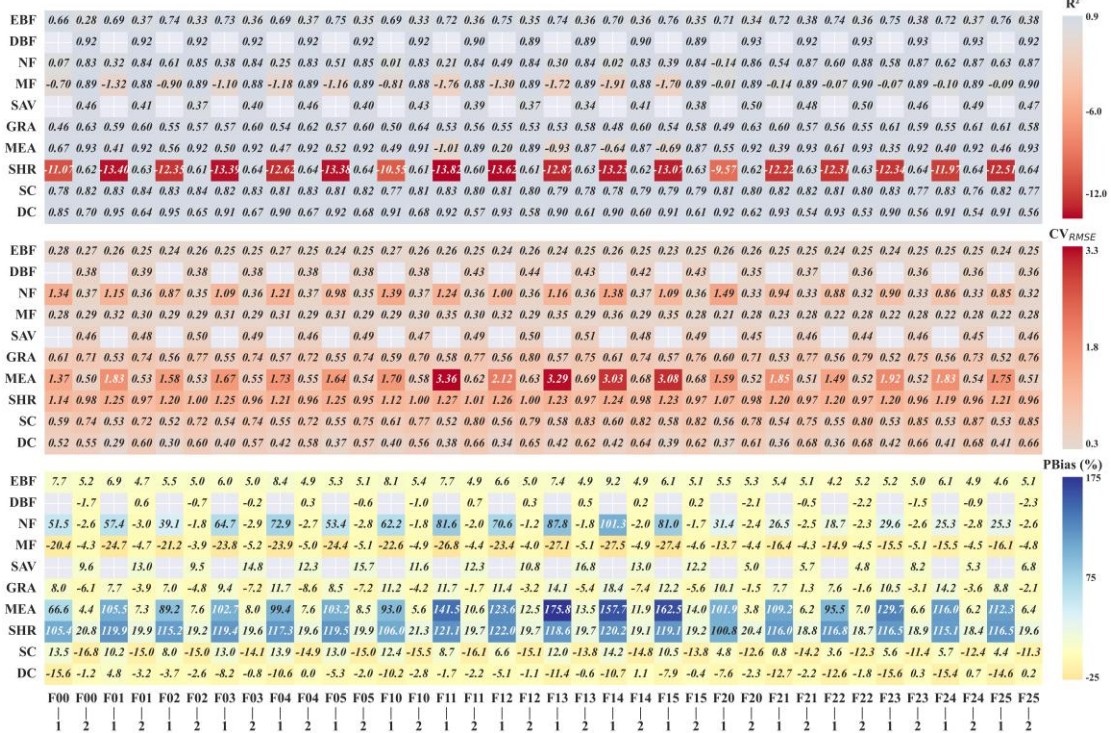

**Figure 12**. Comparison of statistical indicators (R², CV$_{RMSE}$, PBias) of FLAML-LUE model under
drought conditions and non-drought conditions for different land cover types (1 represents
drought, 2 represents non-drought).

## 4. Discussion

Model performance is highly influenced by the algorithms used, the underlying processes, and how GPP responds to varying environmental conditions (Chang et al., 2023). A detailed comparison of the FLAML-LUE models across different ecosystems showed that performance varied depending on the input variables, vegetation types, and time scales (Chang et al., 2023; Harris et al., 2021).

## 4.1 Performance comparison of FLAML-LUE models for different ecosystems

In this study, FLAML-LUE models were constructed for different combinations of variables and different time scales based on AutoML algorithms. On the whole, the modeled GPP values agree well with the GPP estimated based on the EC tower, and the FLAML-LUE models performed better in capturing the magnitude and seasonal dynamics of the GPP, which indicated that it was feasible to estimate the GPP using AutoML algorithms. Further, all three ecosystems showed good model performance driven by observational data. Comparisons across various ecosystems indicate that the model exhibited superior performance over forest ecosystems compared to grassland and agricultural ecosystems, as evidenced by the average $R^2$ values.

Although model performance differences across indicator combinations were minimal, EVI-driven FLAML-LUE models slightly outperformed those driven by NDVI. This highlights the key role of EVI in GPP estimation, as it offers more comprehensive atmospheric correction and is less susceptible to saturation from green reflectance compared to NDVI. Additionally, model performance varied significantly

across sites.
Based on the evaluation metrics, the optimal model selected was FLAML00 (EVI
+ LSWI). Under this combination of indicators, the FLAML-LUE model demonstrated
the best performance in mixed forests at CBF, deciduous broadleaf forests at MEF, and
alpine meadows at HBG_G01, with $R^2$ values of 0.92, 0.92, and 0.93, respectively. The
next best performances were observed in coniferous forests at QYF and HZF, single-
cropping farmland at JZA and SYA stations, double-cropping farmland at YCA, and
typical grasslands at DLG and DMG sites. In contrast, the model performed poorly in
alpine shrub and alpine ecosystems, with an $R^2$ of 0.54, and the worst performance was
observed at the BNF site, with an average $R^2$ of only 0.28. Mixed forests exhibit distinct
seasonal variations that satellite imagery can effectively capture, while evergreen
broadleaf forests (ALF and BNF) show minimal seasonal changes in vegetation cover
or greenness, making accurate predictions challenging. Alpine shrublands have more
complex vegetation structures and less distinct seasonal variations in vegetation cover,
which makes it harder for the model to capture the dynamics accurately. In contrast,
alpine meadows exhibit more pronounced seasonal variations in vegetation cover,
which makes the model more effective in capturing GPP dynamics. For non-forest
ecosystems, the highest $R^2$ values were observed in agricultural fields and typical
grasslands, followed by alpine meadows and alpine shrublands.
Mixed forests display clear seasonal variations that satellite imagery can effectively
capture. However, evergreen broadleaf forests (ALF) have slight seasonal variations in
vegetation cover or greenness, making it difficult for the model to predict. For non-

forest ecosystems, the highest $R^2$ was found in agricultural fields and typical grasslands, followed by alpine meadows and alpine scrub. In addition, the differences in model performance were also reflected in different temporal scales. In general, the model simulation performance at the 16-day and monthly scales was better than that at the daily scale, and the performances of different temporal scales for forest, grassland, and cropland ecosystems were consistent with previous studies.

This study did not distinguish between rainfed and irrigated agricultural systems, considering only the crop rotation types. Specifically, JZA and SYA represent rainfed systems, whereas GCA, LCA, and YCA are irrigated. Future research could incorporate this distinction to improve the accuracy of carbon flux estimates in cropland ecosystems. This distinction is important for interpreting model results under water-limited conditions.

In addition, our results indicate that forest and agricultural fields have greater carbon sequestration capacity and higher annual fluxes than grasslands (Table S9, S10, S11), aligning with previous research outcomes (Y. Wang et al., 2021; Zhang et al., 2007). However, due to the annual harvest of crops, approximately 76% of the on-farm biomass is removed, resulting in limited long-term carbon storage capacity (Zhang et al., 2007). With the exception of tropical rainforests (i.e., BNF), the annual carbon production of planted forests (i.e., QYF) is higher than that of natural forests (i.e., CBF, DHF), which implies that planted forests possess significant potential for carbon assimilation, functioning as robust carbon sinks.

## 4.2 Model Performance Variations Under Extreme Conditions

In the context of global warming and the increasing frequency of extreme climate events, the adaptability and stability of GPP estimation models in extreme environments have become crucial. This study systematically evaluated the performance of the FLAML-LUE model under high-temperature, high-VPD, and drought scenarios by grouping the validation set. The results showed a general decline in the model's accuracy across all three extreme climate conditions, with varying performance depending on the scenario, highlighting the complexity of vegetation carbon absorption responses to climate stress.

In high-temperature conditions, the model generally underestimated GPP. This could be due to the suppression of photosynthesis caused by high temperatures. High temperatures increase transpiration stress, causing stomatal closure to reduce water loss, which limits $CO_2$ input and lowers photosynthetic rates (Qu et al., 2020; Reichstein et al., 2013). Additionally, high temperatures can cause leaf damage and senescence, reducing LAI and overall photosynthetic potential (A. Chen et al., 2021; Y. Chen et al., 2021). Although the FLAML-LUE model accounts for fPAR and water stress factors, it may not fully capture rapid responses such as leaf damage or sudden declines in LAI, which likely contribute to the reduced accuracy under high-temperature conditions. Moreover, the model does not explicitly account for the lag effect of leaf senescence, which may further worsen estimation bias (Frank et al., 2015).

Under high VPD conditions, the model showed significant uncertainty, with some areas overestimating GPP and others underestimating it. This inconsistency likely arises

from the diverse water stress mechanisms induced by high VPD. Guo et al. (2015) noted
that high VPD does not always reflect the true level of water stress in plants, leading to
the potential overestimation of GPP. Conversely, in extreme VPD scenarios, where
stomata close to reduce carbon absorption, the model may underestimate GPP if it fails
to recognize this regulatory behavior (Li et al., 2016). Additionally, the FLAML-LUE
model does not explicitly consider leaf energy load or light inhibition, which may
contribute to the model's higher errors under high VPD conditions (Rigden et al., 2020).

Although the model's performance decreased at some sites under drought

conditions, its overall accuracy improved under these scenarios. This improvement may
be due to the stronger limiting effect of drought on vegetation growth, allowing the
model to more accurately capture the suppressive impact of water stress on
photosynthesis. In drought conditions, water scarcity limits carbon absorption, leading
to a substantial reduction in GPP (McDowell et al., 2008). As a result, the model's
estimates are more likely to align with the actual limitation of carbon absorption. Thus,
under drought conditions, the model may underestimate GPP, which can be more
accurate, while in wetter environments, where water stress is less pronounced, the
model may overestimate GPP, reducing its accuracy. Additionally, under drought, the
model is likely better at capturing the direct effects of water shortage on plant
physiology, reducing interference from other environmental variables and improving
prediction accuracy (Zhou et al., 2019).

Although the FLAML-LUE model demonstrates strong predictive capabilities

under normal climate conditions, there is still room for improvement under extreme

scenarios. One potential limitation is the insufficient representation of rapid plant response mechanisms (e.g., leaf damage and sudden declines in LAI) in the current input features (Frank et al., 2015; Reichstein et al., 2013). Future research could incorporate high-temporal-resolution vegetation indices, such as solar-induced chlorophyll fluorescence (SIF), to better capture dynamic changes in plant metabolic activity and stress responses under extreme conditions (Yi et al., 2024; Pagán et al., 2019). Including lag variables or cumulative stress indices could also enhance the model's ability to handle delayed physiological responses after stress events (Frank et al., 2015). Furthermore, future studies should expand the scope to include a broader range of climate events that affect GPP, such as floods and low temperatures, in addition to high temperature, high VPD, and drought (Wang et al., 2023). Vegetation in different regions responds differently to these events, with low temperatures and frost being especially important for high-latitude ecosystems.

## 4.3 Advantages of FLAML-LUE framework

In this study, FLAML (Wang et al., 2021) selected the Extra Trees algorithm as the best-performing model for GPP simulation in China. Extra Trees is an ensemble learning method that builds multiple unpruned decision trees and incorporates randomization in features selection and split thresholds determination. Compared to traditional decision tree ensembles such as Random Forests, Extra Trees typically achieves minimal variance while maintaining low bias, which makes it particularly well-suited for complex, high-dimensional datasets (Geurts et al., 2006).

The adoption of FLAML provides several significant advantages. First, it

automates the model selection and hyperparameter tuning process, eliminating the need
for extensive manual trial-and-error and reducing reliance on domain expertise (Nakano
and Liu, 2025; Wang et al., 2022). Instead of manually evaluating various algorithms
and their configurations, FLAML efficiently explores a broad search space and
identifies the most appropriate model for the dataset.
Moreover, FLAML employ a cost-aware hyperparameter optimization strategy,
enabling it to find high-performing models with relatively low computational cost
(Zhang et al., 2023; Wang et al., 2021). This feature is particularly advantageous in
scenarios with limited computational resources or the need for rapid prototyping.
Compared to conventional machine learning workflows, FLAML significantly
reduces human bias in model selection, improves reproducibility, and lowers the barrier
to applying advanced modeling techniques (He et al., 2021). Overall, the use of FLAML
in this study not only improved model performance but also streamlined the modeling
process, supporting its broader applicability in ecological and climate-related research.
**4.4 Comparison with other products**

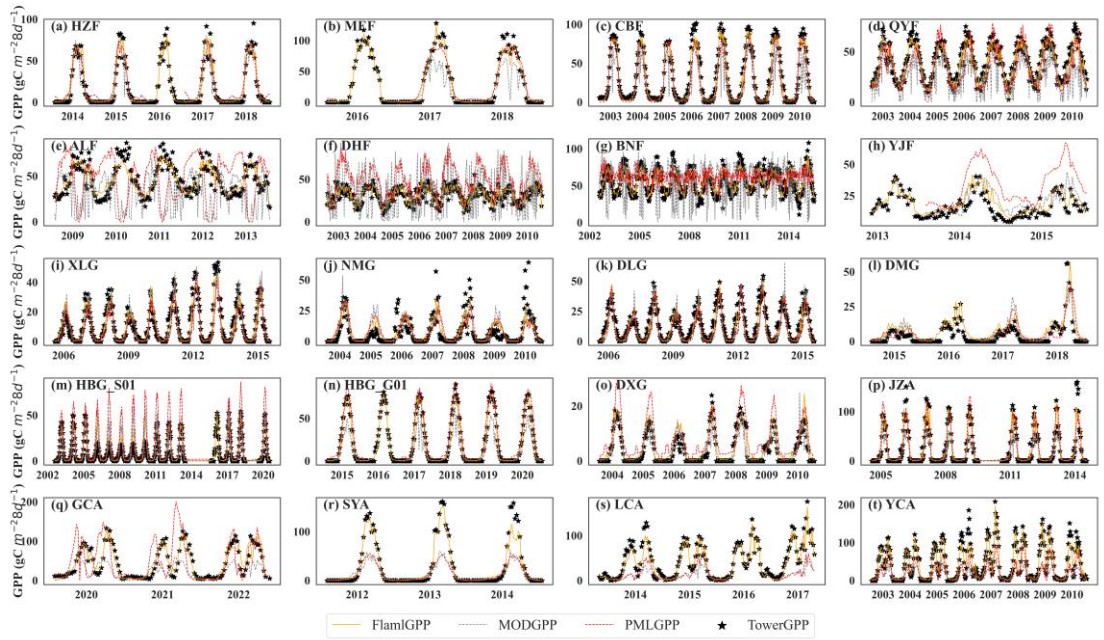


**Figure 15.** Comparing 8-day GPP from FLAML-LUE, PML, MOD17 models, and EC observations.


This study attempted to predict the GPP of different sites using the FLAML model
based on the LUE model variables. The results showed that the AutoML algorithm is a
promising GPP estimation method, which explains on average 75%-98% of the GPP
variation.
Compared to two GPP products (MODIS GPP, and PML GPP), the GPP from this
study showed the highest precision (**Table 7**) and better consistency with flux tower-
based GPP under different ecosystems. Overall, the FLAML-LUE model used in this
study had the best simulation performance. These findings highlight the potential of the
FLAML algorithm for accurately estimating GPP. The FLAML-LUE model is a data-
driven ML approach that builds relationships based on dependent and explanatory
variables. This enables it to effectively simulate the complex nonlinear interactions
across diverse ecosystems (Tramontana et al., 2016). This advantage is even more
prominent at the global scale, considering that more flux tower data are available for
model construction.
**Table 7**
$R^2$ of 8-day GPP simulated by FLAML-LUE, PML, and MOD17 at different ecosystems validation
sites.

| Ecosystem | Station | R2 | | | TSS | | |
|---|---|---|---|---|---|---|---|
| | | FLAML | MOD | PML | FLAML | MOD | PML |
| ALL | ALL | 0.93 | 0.71 | 0.78 | 0.9657 | 0.2677 | 0.5675 |
| Forest | HZF | 0.95 | **0.88** | 0.91 | 0.9843 | **0.9672** | 0.9569 |
| | MEF | **0.98** | 0.78 | **0.95** | 0.9868 | 0.7664 | **0.9571** |
| | CBF | 0.98 | 0.78 | 0.93 | **0.9903** | 0.8860 | 0.9567 |
| | QYF | 0.95 | 0.54 | 0.74 | 0.9833 | 0.8634 | 0.9231 |
| | ALF | 0.87 | 0.24 | 0.34 | 0.9054 | 0.2455 | 0.1812 |
| | DHF | 0.83 | 0.27 | 0.45 | 0.9527 | 0.3030 | 0.5851 |
| | BNF | 0.81 | 0.05 | 0.02 | 0.9025 | 0.3370 | 0.3337 |
| | YJF | 0.75 | 0.31 | 0.42 | 0.9334 | 0.7759 | 0.5820 |
| Grass | XLG | 0.92 | 0.76 | 0.79 | 0.9651 | 0.9343 | 0.9008 |
| | NMG | 0.67 | 0.48 | 0.41 | 0.8288 | 0.8340 | 0.7436 |
| | DLG | 0.92 | 0.76 | 0.77 | 0.9787 | **0.9349** | 0.9320 |
| | DMG | 0.82 | 0.68 | 0.57 | 0.9537 | 0.9080 | 0.8611 |
| | HBG_S01 | 0.89 | 0.78 | 0.81 | 0.9718 | 0.9284 | 0.7175 |
| | HBG_G01 | 0.99 | 0.91 | 0.97 | **0.9947** | 0.7546 | **0.9911** |
| | DXG | 0.90 | 0.75 | 0.82 | 0.9737 | 0.9134 | 0.9105 |
| Crop | JZA | 0.95 | 0.84 | 0.85 | **0.9786** | **0.6009** | **0.9582** |
| | GCA | 0.89 | 0.33 | 0.19 | 0.9708 | 0.4889 | 0.6748 |
| | SYA | **0.96** | **0.92** | **0.92** | 0.9666 | 0.3708 | 0.3948 |
| | LCA | 0.94 | 0.57 | 0.48 | 0.9731 | 0.2433 | 0.3959 |
| | YCA | 0.93 | 0.71 | 0.78 | 0.9657 | 0.2677 | 0.5675 |

Note: Bold numbers indicate the highest values, while underlined numbers represent the lowest
values.
However, further work is needed to evaluate the FLAML-LUE model's suitability
and accuracy, considering its limitations. In particular, it tends to underestimate high
GPP and overestimate low GPP. In addition, the model performance in GPP estimation
is highly dependent on ecosystem type. Our findings indicated that mixed forests,
deciduous broadleaf forests, and agricultural lands had higher prediction accuracies.

While grass sites such as alpine scrub and alpine meadows were predicted with large

uncertainties, consistent with results from other studies (Wang et al., 2021; Yuan et al.,

2014). This is still a big challenge in accurately estimating GPP.

In general, satellite imagery accurately captures the seasonal leaf phenology of

DBF and MF canopies (e.g., spring leaf unfolding and fall senescence). Additionally,

the key environmental factors influencing vegetation production during different

phenological phases are well-defined (Yuan et al., 2014), making them well-suited for

FLAML-LUE modeling. In contrast, the ambiguous seasonal leaf area changes in EBF

and the low variability of GPP in NMG ecosystems result in poorer model performance,

and empirical methods struggle to estimate GPP variability in these areas (Tramontana

et al., 2016).

Model performance is heavily influenced by the quality of the driver data and the

typicality of the flux towers. In this study, meteorological indices are obtained directly

from spatially explicit reanalysis products. Remotely sensed variables (e.g., NDVI and

EVI, LSWI) serve as proxies for vegetation growth and seasonal changes and are crucial

for scaling simulations from site to regional levels. These gridded indices are directly

derived from satellite reflectance bands. Large-area EFs can be obtained using LE and

Hs calculations from ERA5 reanalysis data or can be derived using NDVI temperature

triangulation (Venturini et al., 2004). LAI, VPD, Pre, and RH can be obtained from

ERA5 reanalysis data. Thus, the model can be extended from the site scale to the

regional and even global scale. Building on this foundation, we will develop a long-

term gridded GPP dataset for China using the FLAML-LUE framework to analyze its

spatiotemporal variations over multiple years. This dataset will allow us to investigate long-term GPP trends across different climate zones and vegetation types, as well as their responses to key environmental drivers. By comparing GPP estimates across regions and years, we will also assess model uncertainties and identify potential areas for improvement.

## 5. Conclusion

In this study, the FLAML-LUE model was developed based on data from 20 flux observation sites across China, integrating the FLAML algorithm with key variables from the LUE model. The results demonstrate that the FLAML-LUE model performs excellently in GPP prediction, accurately simulating both its temporal variations and magnitude, particularly in mixed forests and coniferous forests. The average $R^2$ for daily-scale simulations reached 0.92 and 0.91, respectively. Further analysis showed that extending the temporal scale of input data significantly improves model accuracy. In a comparison of models with different variable combinations, it was found that the model driven by EVI outperformed those driven by NDVI and LAI. The model using LSWI as the driving variable performed better than those with EF, SW, CPD, Pre, and RH as primary variables, with the EVI+LSWI combination yielding the best performance. Additionally, the model's prediction accuracy decreased under high temperature and high VPD conditions. However, under drought conditions, the overall prediction accuracy increased, although it decreased at some sites.

In summary, the FLAML-LUE model demonstrates strong applicability and potential for wider application in GPP estimation. It holds promise for scaling from site-

level to regional or even global levels, contributing to a deeper understanding of carbon
cycling processes. However, the model's applicability in unique ecosystems, such as
alpine shrublands, remains limited, and its ability to adapt to extreme climate events
requires further enhancement. Future work should focus on optimizing the model
structure and parameter settings to improve its robustness and generalization across
diverse ecological environments.
## CRediT authorship contribution statement
J.L., Y.Z. and J.W. conceived the study. J.L. collected and processed the data. J.L.
and Y.Z. drafted the manuscript. A.W., Y.Z., R.L and W.D. funded the study. J.L., Y.Z.,
A.W, W.F. and J.W. checked the negatives and touched up. All authors have read and
agreed to the embellished manuscript.
## Data availability
A Fast Library for Automated Machine Learning & Tuning (FLAML) is a Python
library, and detailed documentation about FLAML can be found on GitHub. We have
uploaded the related source code and documentation to Zenodo
(https://doi.org/10.5281/zenodo.14874754, Laijie, 2025). The flux observation data and
the Python source code of the FLAML-LUE used in this paper are also archived on
Zenodo (https://doi.org/10.5281/zenodo.15477703, Laijie, 2024).
## Declaration of competing interests
The authors declare that they have no known competing financial interests or
personal relationships that could have appeared to influence the work reported in this
paper.

## Acknowledgments

This study was financially supported by the National Key Research and Development Program of China (Grant Number: 2022YFF1300501), the Natural Science Foundation of Liaoning Province (Grant Number: 2024-BSBA-62), the Open Research Fund Project of Key Laboratory of Ecosystem Carbon Source and Sink, China Meteorological Administration (Grant Number: ECSS-CMA202305), the Fundamental Research Funds of the Chinese Academy of Meteorological Sciences (Grant Number: 2024Z001). This work utilized eddy covariance data obtained from ChinaFlux. We appreciate all the staff at ChinaFlux for providing high-quality measurement data to the scientific community.

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
