# Peer review of "FLAML version 2.3.3 model-based assessment of gross"

_Geoscientific Model Development, 2024_

## Author Comment (AC1)

Response

1、 We have to ask you to store the FLAML code and all the data that you use to produce your manuscript in one of the acceptable repositories according to our policy.

Response: It seems there may have been a misunderstanding regarding this section. As per the journal's policy, we have stored both our code and data on Zenodo (DOI: 10.5281/zenodo.14542880). The GitHub link provided in our manuscript refers to the FLAML source code, which is maintained by its developers and is included solely to facilitate access to the library for other researchers.

The site-level flux data were obtained from https://www.scidb.cn/en/. However, the actual dataset used in our study has been processed and is also available on Zenodo.

If you believe our Code and Data Availability section still does not fully comply with the journal's guidelines, we would be happy to revise it accordingly. We look forward to your feedback.

---

## Author Comment (AC2)

Response

1、Revisions to the Code and Data Availability section.

Response: We hope this message finds you well. Thank you very much for your valuable feedback. We have carefully considered your suggestions and made the necessary revisions. The FLAML source code in a zip file has been uploaded to Zenodo.

With the modifications outlined below, we believe it now fully complies with the requirements of the Code and Data Availability section.

"A Fast Library for Automated Machine Learning & Tuning (FLAML) is a Python library, and detailed documentation about FLAML can be found on GitHub. We have uploaded the related source code and documentation to Zenodo (https://doi.org/10.5281/zenodo.14874754, Laijie, 2025). The flux observation data and the Python source code of the FLAML-LUE used in this paper are also archived on Zenodo (https://doi.org/10.5281/zenodo.14542880, Laijie, 2024). "

---

## Author Comment (AC3)

Response

Q1. Line 67: There is a period before the citation and another one after it. Please pay attention to this detail error.

Thank you for pointing out this detail error. We apologize for the oversight regarding the duplicate periods before and after the citation in Line 67. This has now been corrected in the revised manuscript. We appreciate your careful review and valuable feedback, which has helped improve the quality of our paper.

Q2. Line 143: The authors should consider including the geographical coordinates (longitude and latitude) of the monitoring stations in Table 1. Providing this spatial reference would significantly enhance the study's reproducibility and facilitate comparative analyses with other datasets.

We have now included the geographical coordinates (longitude and latitude) of the monitoring stations in Table 1, as recommended. This addition enhances the study's reproducibility and facilitates comparative analyses with other datasets.

Q3. Line 147 and 150: It is recommended to change the last third of the sentence to: "The last third of the long-term series data from the ALF, CBF, and QYF stations were used for forest model validation. Similarly, the last third of the data from DLG, DXG, and HBG stations were used for grassland model validation, and the last third of the data from JZA and YCA stations were used for cropland model validation."

We appreciate your suggestion to improve the clarity of the sentence. Following your recommendation, we have revised the sentence to: "The last third of the long-term series data from the ALF, CBF, and QYF stations were used for forest model validation. Similarly, the last third of the data from DLG, DXG, and HBG stations were used for grassland model validation, and the last third of the data from JZA and YCA stations were used for cropland model validation."

Q4. Line 156: It is recommended to change "However, some sites have no ER data" to "some sites lacked ER data."

We have revised the sentence to: "some sites lacked ER data," as suggested, to improve conciseness and clarity.

Q5. Line 216 and 219: (C. Wang et al., 2021) should be (Wang et al., 2021).

We apologize for the citation format error. The citation has been corrected to (Wang et al., 2021) in Lines 216 and 219, as per your recommendation.

Q6. Line 241: The table title does not have a period at the end.

Thank you for catching this oversight. We have added a period at the end of the table title in Line 241 to ensure proper formatting.

Q7. Line 256-259: All equations should be centered in the text, with equation numbers placed on the right margin enclosed in parentheses.

We appreciate your attention to detail. All equations have now been centered in the text, and equation numbers have been placed on the right margin enclosed in parentheses, as recommended.

Q8. Line 272: The manuscript exhibits inconsistent punctuation formatting patterns that require standardization.

Thank you for highlighting this issue. We have carefully reviewed the manuscript and standardized the punctuation formatting patterns to ensure consistency throughout the text.

Q9. In Figures 4, 8, and 12: The y-axis labels appear to be incorrect. Shouldn't they be 8-day, 16-day, and monthly, respectively?

Thank you for your careful review. We have corrected the y-axis labels in Figures 4, 8, and 12 to '8-day,' '16-day,' and 'monthly,' as suggested.

Q10.Line 430, 452, and 545: The 2 in $R^2$ is not formatted as a superscript.

Thank you for pointing out this formatting issue. We have corrected the superscript formatting for $R^2$ in Lines 430, 452, and 545.

Q11.Why are separate models constructed for forest, grassland, and cropland ecosystems instead of developing a single unified model?

Thank you for your insightful question. Separate models were developed for forest, grassland, and cropland ecosystems due to the substantial differences in their biophysical characteristics, vegetation dynamics, and environmental responses. A single unified model would struggle to accurately capture these ecosystem-specific variations, potentially leading to reduced predictive accuracy and applicability.

In our study, we found distinct differences in the key factors influencing GPP across ecosystems. In forest ecosystems, EVI performed best among vegetation indices, while LSWI was the most effective moisture index. However, moisture indices had relatively low overall importance compared to other variables. In grassland ecosystems, EVI, NDVI, and LAI exhibited similar performance, with LSWI emerging as the most influential moisture index. In contrast, cropland ecosystems were characterized by LAI as the most important vegetation index, while moisture indices played a crucial role, ranking just behind temperature and vegetation indices. Notably, the primary moisture factor affecting GPP simulation in croplands was LSWI, followed by PDSI, with EF contributing the least.

Given these ecosystem-specific variations, using a single model would risk oversimplifying the complex interactions between environmental factors and vegetation responses. By constructing separate models, we ensure that each ecosystem's unique characteristics are properly represented, thereby enhancing the robustness, accuracy, and ecological relevance of our findings.

---

## Author Comment (AC4)

**General Comments**

Q1. First, the choice of simple vegetation indices as dependent variables for the model seem to me dated, especially due to the current availability of Solar Induced Fluorescence (SIF) products, which are more suited as proxies of photosynthesis than EVI, NDVI, etc. Although the authors mention the possible future use of SIF, I would like to know further details to why it was not used in this study, or extra analysis where SIF is included.

Thank you for your insightful comment. We acknowledge that Solar Induced Fluorescence (SIF) is a promising proxy for photosynthesis and has been increasingly used in recent studies. Compared to traditional vegetation indices (e.g., EVI, NDVI), SIF directly reflects chlorophyll fluorescence emissions, providing a more direct link to gross primary production (GPP).

However, in this study, we did not incorporate SIF due to the following reasons:

Data Availability: Solar-induced fluorescence (SIF) observations have significantly advanced in recent years, yet the availability of long-term, continuous SIF datasets with fine spatial resolution remains a challenge. In comparison to well-established vegetation indices such as the Enhanced Vegetation Index (EVI) and the Normalized Difference Vegetation Index (NDVI), which have been monitored for decades using sensors like MODIS, SIF datasets are relatively recent. The SIF data listed in Table 1 highlight various datasets with different temporal coverage, spatial resolutions, and geographic extents. While some datasets, such as GOME-2 and OCO-2, provide global coverage and span several years, none of the available datasets fully meet the temporal coverage requirements for all FLUX station periods. Additionally, combining SIF products from different sources could introduce inconsistencies, leading to potential errors. These inconsistencies pose a significant risk to the reliability and accuracy of analyses, which is why we chose not to use these SIF products for generating a long-term time series.

**Table 1** Summary of Satellite Datasets for Solar-Induced Fluorescence (SIF) Observations.

| Dataset | Temporal coverage | Spatial resolutions | Time resolutions | Coverage |
| --- | --- | --- | --- | --- |
| GOME-2 | 2007 to present | 40 km $\times$ 40 km | 1-2 days | Global |
| OCO-2 | 2014 to present | 1.3 km $\times$ 2.25 km | 16 days | Global |
| TROPOMI | 2018 to present | 7 km $\times$ 7 km | 1 day | Global |
| GOSAT | 2009 to present | 10 km $\times$ 10 km | 3 days | Global |
| SCIAMACHY | 2002－2012 | 30 km $\times$ 60 km | 35 days | Global |
| TanSat | 2016 to present | 1 km $\times$ 2 km | 16 days | Global |
| OCO-3 | 2019 to present | 1.6 km $\times$ 2.2 km | 16 days | Global |
| CFIS | 2016－2018 | 30 m $\times$ 30 m | Irregular | Local |
| TANSO-FTS | 2009 to present | 10 km $\times$ 10 km | 3 days | Global |

Resolution Limitations: The current global SIF products, such as those from OCO-2 and TROPOMI, often have spatial resolutions that are relatively coarse, typically greater than 1 km. While suitable for large-scale or global studies, this level of resolution is insufficient for capturing fine-scale ecological variations, particularly

in heterogeneous or fragmented landscapes. For instance, OCO-2's spatial resolution of 1.3 km $\times$ 2.25 km and TROPOMI's 7 km $\times$ 7 km resolution may not be ideal for studies requiring detailed local information or the monitoring of small-scale ecosystem dynamics. Some datasets like CFIS, with a resolution of 30 m $\times$ 30 m, offer much finer spatial detail while their spatial coverage of datasets is usually incomplete, which cannot meet our continuous and full flux sites coverage needs in a large area.

For these reasons, we did not incorporate SIF datasets in our current study. That being said, we acknowledge the potential benefits of incorporating SIF and are considering its integration in future research. We plan to explore whether SIF-based models can further improve GPP estimations, either as a standalone predictor or in combination with traditional vegetation indices. Once again, we appreciate your valuable suggestion and will take this into account in our future work.

Q2. Second, the resolution of the remote sensing products used (500 meters) does not seem to be compatible with the eddy flux data. At this scale, microclimatic or topographic factors may cause significant divergences in relation to a 500 m size pixel, and lead to inconsistencies. I suggest that if possible data with higher resolution are used (LANDSAT or SENTINEL-2) or arguments are given for the use of the lower resolution product.

Thank you for your thoughtful suggestion regarding the spatial resolution of the remote sensing products used in our study.

First, we understand your concern that the 500 m spatial resolution of MODIS data might not be ideal for capturing fine-scale variations relevant to eddy covariance measurements. However, it is important to note, as described by Schmid (2002), that the footprint of an eddy covariance tower is not fixed but varies with meteorological conditions, typically ranging from 100 m to 1 km. Additionally, Zhang et al. (2021) found that different footprints, such as 500, 1000, and 1500 meters, showed almost no difference in the study area. Given this, we believe that the 500 m resolution of MODIS is appropriate for representing the footprint of the flux tower and is well-suited for our study.

We did consider the use of higher-resolution products, such as LANDSAT and SENTINEL-2, but there are a few important limitations associated with these datasets.

Regarding LANDSAT data, although it offers finer spatial resolution, there are known issues with data quality. Several Landsat satellites, including Landsat 7, suffered from technical failures that resulted in data gaps and missing information. These issues compromise the consistency and reliability of the dataset, particularly for long-term monitoring studies. As a result, the data quality and temporal consistency of LANDSAT may not be suitable for this study.

As for SENTINEL-2, although it provides high-resolution imagery (10 m), its temporal coverage is limited compared to MODIS. SENTINEL-2 data is available since 2015, which means it doesn't fully cover the historical periods needed for our analysis, especially for longer-term studies. Furthermore, while SENTINEL-2 offers good spatial resolution, it may not always be available due to cloud cover and other

environmental factors, further complicating its use for continuous monitoring.

Considering these limitations, we chose to use MODIS data with 500 m resolution because it offers a good balance between spatial resolution, temporal coverage, and global availability, making it more suitable for our study's long-term monitoring needs.

We hope this clarifies our choice of data and addresses your concerns. Thank you again for your valuable input, which will help us refine our approach.

Q3. Finally, I would be very interested in the production of a GPP map of China using the FLAML framework, and how it compares with other GPP maps. I think this would greatly increase the manuscript's appeal.

Thank you for your valuable suggestion. Your input has provided us with very useful inspiration. Using the FLAML framework to create a GPP (Gross Primary Productivity) map for China is indeed a meaningful and interesting task. As we have mentioned in the text, the FLAML-LUE models have "the potential to be applied in predicting GPP for different vegetation types at a regional scale". However, these models are only driven from data of 20 stations, which is not enough to cover the entire ecosystem types in China. Therefore, using them for the production of a China GPP map is still not competent enough. This is not related to the limitations of the method, it's just that we need more site data support.

We plan to further develop this aspect in our future research and will provide a detailed discussion of it in the manuscript. We will consider using the FLAML framework to build a GPP prediction model for China and compare it with existing GPP maps to assess its accuracy and applicability. This will not only help us better understand the spatial distribution of GPP in China but also provide valuable insights for global GPP research.

Once again, thank you for your insightful feedback. Your suggestion will undoubtedly enrich the depth and scope of our research. We will continue to explore this direction in our future work and present the results more comprehensively in the manuscript.

**Specific comments**

Q1. L90 - I would not say ML is "fundamentally different" from regression models, but that they offer advantages in relation to.

Thank you for your insightful comment. You are absolutely right that machine learning is not fundamentally different from regression models but rather offers advantages in certain aspects. We have revised the text accordingly to better reflect this distinction. The revised sentence now reads: "ML is a modeling approach that differs from simple regression models and complex simulation models in its methodology."

Q2. L94 - I would also point out limitations on ML techniques, such as dependence on large training datasets and not being able to link results to real-world processes.

Q3. L96 - ...Which is an advantage when the focus is solely on spatial predictions

Response to Q2 (L94) and Q3 (L96): Thank you for your valuable comments. We acknowledge that machine learning techniques have certain limitations, including their dependence on large training datasets and the challenge of directly linking results to real-world processes. These constraints are important considerations when applying ML models. However, as you pointed out, when the primary focus is on spatial predictions, the ability of ML models to capture complex patterns without requiring explicit process-based formulations can be an advantage. We have revised the manuscript to reflect these points more clearly.

We have revised our manuscript as follows:

"These data-driven models are particularly suited for capturing nonlinear ecosystem dynamics but often require large training datasets and may lack explicit links to real-world processes. However, their ability to uncover spatial patterns without process-based constraints makes them valuable for spatial predictions. Consequently, ML-based approaches have gained popularity in recent years. For example, Kong et al. (2023) developed a hybrid model that combines ML and LUE model to estimate GPP. This hybrid model improves the LUE model by integrating a machine learning approach (MLP, multi-layer perceptron), and estimates GPP using the MLP-based LUE framework along with additional required inputs."

Q4. Fig. 1 - The mini-map on the bottom right corner does not include any sites, or any extra information, maybe remove it? Otherwise, I believe the editors should label these areas in the South China Sea as "under dispute", as stated in the "maps and aerials" section of the submission guidelines.

We sincerely thank the reviewers for their valuable suggestions regarding the mini-map in Figure 1. However, we would like to clarify that the map reflects the distribution of flux sites within China's territory. As required, we have ensured that the map accurately represents China's territorial boundaries. This representation is consistent with the practices in previous publications in Geoscientific Model Development (GMD). For example, in the article by Ren et al. (Ren et al., 2024), Figure 1, and in Figure 1 of the article by Wang et al. (Wang et al., 2022) and Figure 2 of the article by Wu et al. (Wu et al., 2021) , the South China Sea is similarly depicted as part of China's territory without any additional labels indicating disputes.

We understand the sensitivity of territorial issues and the importance of adhering to journal guidelines. However, given the scientific context of our study and the precedent set by other publications in GMD, we believe that the current representation of the map is appropriate. We hope this explanation addresses the reviewer's concern.

Q5. Table 2 – In contrast to other vegetation indexes, LAI satellite data is based on empirical models, such as previous GPP estimating methods. It would be interesting to check if field LAI data from the sites are available to see if direct LAI measurements improve the ML model.

Thank you for your insightful comment. You are absolutely right that LAI satellite data, unlike other vegetation indices, is often derived from empirical models,

similar to GPP estimation methods. We appreciate your suggestion to explore the availability of field LAI data from the study sites. We also believe that incorporating direct field LAI measurements could potentially enhance the performance of the ML model by providing more accurate and site-specific information. Unfortunately, at this stage, field LAI data in most of the 20 sites were not available. However, we plan to explore this avenue in future research and will certainly consider integrating field measurements of LAI if they become available, as they may provide valuable improvements to the model.

Q6. L686 - I would argue then that in the future hyperspectral data + ML would provide much better estimates too, this could be discussed with references.

Thank you for your valuable suggestion. We agree that hyperspectral data, when combined with machine learning (ML) techniques, could provide more accurate and robust estimates in the future. Hyperspectral data offer a rich spectrum of information across many wavelengths, which can capture subtle variations in vegetation properties that other remote sensing datasets might miss. This could indeed improve model predictions by providing more detailed spectral features.

We have revised our manuscript as follows:

"Recent research indicates that satellite observations of solar-induced chlorophyll fluorescence (SIF) provide a more accurate picture of plant photosynthesis dynamics and serve as a more effective indicator for modeling subtropical evergreen vegetation (Sun et al., 2017; Frankenberg et al., 2011). In the future, integrating hyperspectral data with machine learning could lead to more accurate GPP estimates, as hyperspectral data offer finer spectral resolution, enabling better capture of vegetation traits and environmental conditions (Gessner et al., 2015; Zarco-Tejada et al., 2013). This integration could further enhance model performance, particularly for evergreen forests. For example, Zhang et al. (2021) used hyperspectral data (EO-1 Hyperion) to estimate GPP in the temperate forests of Changbai Mountain. Future research should consider incorporating both hyperspectral data and SIF into models to assess their potential for improving GPP estimations across various ecosystems."

We appreciate your input and will explore the literature on this topic to strengthen our discussion.

**Reference**

Frankenberg, C., Fisher, J.B., Worden, J., Badgley, G., Saatchi, S.S., Lee, J.-E., Toon, G.C., Butz, A., Jung, M., Kuze, A., Yokota, T., 2011. New global observations of the terrestrial carbon cycle from GOSAT: Patterns of plant fluorescence with gross primary productivity. Geophys. Res. Lett. 38. https://doi.org/10.1029/2011GL048738

Gessner, U., Machwitz, M., Esch, T., Tillack, A., Naeimi, V., Kuenzer, C., Dech, S., 2015. Multi-sensor mapping of West African land cover using MODIS, ASAR and TanDEM-X/TerraSAR-X data. Remote Sens. Environ. 164, 282–297. https://doi.org/10.1016/j.rse.2015.03.029

Kong, D., Yuan, D., Li, H., Zhang, J., Yang, S., Li, Y., Bai, Y., Zhang, S., 2023. Improving the

Estimation of Gross Primary Productivity across Global Biomes by Modeling Light Use Efficiency through Machine Learning. Remote Sens. 15, 2086. https://doi.org/10.3390/rs15082086

Ren, F., Lin, J., Xu, C., Adeniran, J.A., Wang, J., Martin, R.V., van Donkelaar, A., Hammer, M.S., Horowitz, L.W., Turnock, S.T., Oshima, N., Zhang, J., Bauer, S., Tsigaridis, K., Seland, Ø., Nabat, P., Neubauer, D., Strand, G., van Noije, T., Le Sager, P., Takemura, T., 2024. Evaluation of CMIP6 model simulations of $PM_{2.5}$ and its components over China. Geosci. Model Dev. 17, 4821–4836. https://doi.org/10.5194/gmd-17-4821-2024

Schmid, H.P., 2002. Footprint modeling for vegetation atmosphere exchange studies: a review and perspective. Agric. For. Meteorol., FLUXNET 2000 Synthesis 113, 159–183. https://doi.org/10.1016/S0168-1923(02)00107-7

Sun, Y., Frankenberg, C., Wood, J.D., Schimel, D.S., Jung, M., Guanter, L., Drewry, D.T., Verma, M., Porcar-Castell, A., Griffis, T.J., Gu, L., Magney, T.S., Köhler, P., Evans, B., Yuen, K., 2017. OCO-2 advances photosynthesis observation from space via solar-induced chlorophyll fluorescence. Science 358, eaam5747. https://doi.org/10.1126/science.aam5747

Wang, P., Mao, K., Meng, F., Qin, Z., Fang, S., Bateni, S.M., 2022. A daily highest air temperature estimation method and spatial–temporal changes analysis of high temperature in China from 1979 to 2018. Geosci. Model Dev. 15, 6059–6083. https://doi.org/10.5194/gmd-15-6059-2022

Wu, R., Tessum, C.W., Zhang, Y., Hong, C., Zheng, Y., Qin, X., Liu, S., Zhang, Q., 2021. Reduced-complexity air quality intervention modeling over China: the development of InMAPv1.6.1-China and a comparison with CMAQv5.2. Geosci. Model Dev. 14, 7621–7638. https://doi.org/10.5194/gmd-14-7621-2021

Zarco-Tejada, P.J., Guillén-Climent, M.L., Hernández-Clemente, R., Catalina, A., González, M.R., Martín, P., 2013. Estimating leaf carotenoid content in vineyards using high resolution hyperspectral imagery acquired from an unmanned aerial vehicle (UAV). Agric. For. Meteorol. 171–172, 281–294. https://doi.org/10.1016/j.agrformet.2012.12.013

Zhang, Y., Wang, A., Yuan, F., Guan, D., Wu, J., 2021. The application of EO-1 Hyperion hyperspectral data to estimate the GPP of temperate forest in Changbai Mountain, Northeast China. Environ. Earth Sci. 80, 353. https://doi.org/10.1007/s12665-021-09639-x

---

## Author Comment (AC5)

**Response to Anonymous Referee #2 (https://doi.org/10.5194/gmd-2024-169-**

**RC3)**

The authors apply FLAML v2.3.3---an automated machine-learning toolkit---to predict gross primary productivity (GPP) across 20 eddy-covariance sites, which is a less interesting and less novel endeavor. The manuscript would benefit from a more sharply defined research question and a deeper interrogation of the ecological processes underlying the model's performance. In particular, the authors should clarify what novel scientific insight they seek --- beyond demonstrating sensitivities --- and explore how specific feature groups/selections inform mechanistic understanding rather than merely reflecting data redundancy and uncertainty. Given these substantive concerns about framing and ecological interpretation, I respectfully decline to continue with further review, if that is the case.

We sincerely appreciate your valuable time and insightful comments, which have significantly helped us improve the quality and clarity of our manuscript. In the revised version, we have carefully addressed all the issues you raised. Specifically, we have thoroughly revised the structure and content of the manuscript, resulting in substantial modifications—nearly a thousand changes were made throughout the document.

We believe that these revisions have greatly strengthened the overall presentation and scientific value of our work. Below, we provide a detailed point-by-point response to each of your comments.

**General Comments**

Q1. The whole work reads more like a sensitivity report than an ecological modeling study. What specific scientific insight are the authors seeking by comparing FLAML to not scientifically different feature groups?

Thank you for your thoughtful and constructive comment. We have further clarified the scientific rationale and objectives of our study in the Introduction section of the manuscript. Our study aims to bridge the gap between process-based ecological modeling and data-driven approaches by integrating domain-specific knowledge from LUE models with the automated and efficient learning capabilities of FLAML. The resulting FLAML-LUE framework is a knowledge-guided machine learning model designed to address key ecological questions related to the estimation of GPP.

Specifically, our scientific insights are centered on the following (Line 122-131):

- To evaluate the performance of models using different combinations of LUErelated variables, such as absorbed PAR (fPAR) and water stress factors, across multiple vegetation types and time scales.
- To investigate model robustness under extreme climatic conditions, including high temperatures, elevated vapor pressure deficits (VPD), and drought. By evaluating model stability under these stressors, we aim to assess the resilience and reliability of GPP estimation frameworks in the face of climate variability and change.

The ultimate objective is to identify optimal input combinations for FLAML-LUE

models tailored to different vegetation types and climate zones across China. This helps enhance regional-scale GPP estimation accuracy, which is crucial for carbon budget assessments and ecosystem management.

Q2. The main text suggests a "FLAML-LUE model", yet none of the analyses explicitly implement light-use-efficiency (LUE) theory. Instead, all results derive from various tree-based regressors. If the intent is to compare FLAML-derived machine-learning models against LUE theory, the authors should at least incorporate an explicit LUE model.

Thank you for your thoughtful comment. We have further clarified the structural framework of the FLAML-LUE model in Section 2.3.3 of the manuscript (Lines 122 and 272). In this study, the term "FLAML-LUE" does not refer to a direct implementation of a mechanistic light-use efficiency (LUE) model. Rather, it reflects a hybrid modeling strategy where we incorporate key explanatory variables that originate from LUE theory—such as absorbed photosynthetically active radiation (fPAR), light-use efficiency modifiers, and environmental stress indicators (e.g., VPD, temperature, and water stress indices) — into an automated machine learning framework (FLAML). These variables represent the core components influencing vegetation productivity in traditional LUE models.

$$GPP = f (PAR, T, fPAR, W_j, VT, Season)$$
(3)

where, the *fPAR* include EVI, NDVI, and LAI;  $W_j$  denotes moisture factors including LSWI, EF, SW, PDSI, Pre, RH; *VT* represents vegetation types, in which forest ecosystems include: EBF, DBF, NF, MF, and SAV; grassland ecosystems include GRA, MEA, and SHR, and farmland ecosystems include SC and DC; *Season* represents the season in which the original data were acquired.

Our goal was to combine domain knowledge from LUE theory with the flexibility and efficiency of data-driven models. While we do not simulate GPP using a process-based LUE equation, the LUE-related predictors guide the learning process of the machine learning models, enabling a knowledge-informed estimation of GPP across different vegetation types and environmental conditions.

Q3. The model groups differ mainly in dryness index definition, data source or temporal averaging (e.g., PDSI vs. evaporative fraction, flux - tower vs. ERA5-Land temperature, actually Ta\_flux is typically gapfilled by ERA5). These inputs often carry overlapping information, so comparisons may reflect data uncertainty or scale mismatches rather than mechanistic differences. Exploring a truly critical predictor --- such as soil moisture --- could strengthen the ecological relevance and offer interesting insights. A basic clarification to mention here is that ERA5-Land is a reanalysis dataset rather than a remote sensing product, and it should not be confused with ERA5. ERA5Land provides hourly rather than daily data.

Thank you for your valuable suggestion. We have addressed both issues you raised with corresponding revisions.

First, based on your comments, we have revised the selection of input variables used

in the model construction process (see **Table 1**). Following this adjustment, we retrain the models and re-evaluated the results accordingly. Specifically, to ensure consistency and reliability across all 18 variable combinations, we standardized the sources of temperature and PAR data by uniformly adopting ERA5-Land products. Additionally, we removed the PDSI dataset from our analysis because it is only available at a monthly temporal resolution, which is inconsistent with the finer time scales of other datasets used in this study. Instead, we carefully selected variables that more accurately capture vegetation moisture constraints from multiple ecological perspectives: atmospheric moisture stress (e.g., relative humidity and precipitation), vegetation-level moisture stress (e.g., LSWI and EF), and soil moisture limitations (e.g., SW). These choices are grounded in ecological theory and supported by previous research (Chang et al., 2023).

| - |   |    |     |  |
|---|---|----|-----|--|
|   | 6 | h  |     |  |
| л | a | U. | IC. |  |
|   |   |    |     |  |

| Group   | Input variables | Group   | Input variables | Group   | Input variables |
|---------|-----------------|---------|-----------------|---------|-----------------|
| FLAML00 | NDVI, LSWI      | FLAML10 | EVI, LSWI       | FLAML20 | LAI, LSWI       |
| FLAML01 | NDVI, EF        | FLAML11 | EVI, EF         | FLAML21 | LAI, EF         |
| FLAML02 | NDVI, SW        | FLAML12 | EVI, SW         | FLAML22 | LAI, SW         |
| FLAML03 | NDVI, VPD       | FLAML13 | EVI, VPD        | FLAML23 | LAI, VPD        |
| FLAML04 | NDVI, Pre       | FLAML14 | EVI, Pre        | FLAML24 | LAI, Pre        |
| FLAML05 | NDVI, RH        | FLAML15 | EVI, RH         | FLAML25 | LAI, RH         |

Input variable combinations of fPAR and water stress indicators.

Regarding the second issue you mentioned about the description of the ERA5-Land dataset, we have made corresponding revisions in the updated manuscript. Specifically, **Section 2.2.3** now reads as follows: "ERA5-Land (Hersbach et al., 2020) is a global high-resolution reanalysis dataset produced by the European Centre for Medium-Range Weather Forecasts (ECMWF) under the Copernicus Climate Change Service (C3S). It provides hourly land surface variables at a spatial resolution of 0.1°, generated using a dedicated land surface model driven by the ERA5 climate reanalysis. The dataset integrates advanced land surface modeling and data assimilation techniques, offering a wide range of variables such as air temperature, soil moisture, precipitation, and snow depth. In this study, site-specific variables including air temperature (T), soil water content (SW), precipitation (Pre), and leaf area index (LAI) were extracted from ERA5-Land. In addition, photosynthetically active radiation (PAR), evapotranspiration fraction (EF), VPD and relative humidity (RH) were calculated and derived from available ERA5-Land variables using GEE."

Once again, thank you for your insightful feedback. Your suggestions have significantly contributed to improving the depth and rigor of our study. We will continue to build on this work and aim to present our findings more comprehensively in future research.

Q4. The rationale for analyzing 8-day, 16-day vs. monthly statistics is not fully

developed. Because GPP seasonality dominates many signals, the differences in model performance may simply reflect sample size (it is unsurprising that monthly R2 exceed those at the 8-day scale, and this comparison offers no insight).

Thank you for your insightful comment. We agree that the seasonal dynamics of GPP and the differences in sample sizes across temporal scales (e.g., 8-day, 16-day, monthly) can inherently influence model performance metrics such as  $R^2$ . However, our rationale for analyzing multiple temporal resolutions goes beyond statistical comparisons.

The primary objective of incorporating different temporal scales is to evaluate the robustness and generalizability of the FLAML-LUE model across varying degrees of temporal aggregation. As indicated in the revised manuscript (Line 464 - 467), compared to the daily scale, the nuRMSE decreases by 12.97%, 16.52%, and 25.92% at the 8-day, 16-day, and monthly scales, respectively. This highlights that the uncertainty of the FLAML-LUE model is significantly reduced at coarser temporal resolutions.

Furthermore, from an application perspective, transitioning from site-level to regional-scale GPP estimation across China requires temporal resolutions that align with commonly used satellite products. In this context, 8-day or monthly models are more practical, as they not only reduce noise through temporal aggregation but also ensure greater consistency with large-scale remote sensing data. These coarser time scales offer a more effective trade-off between capturing ecological dynamics and enabling broader spatial applicability.

**Q5. Presenting each PFT group in separate sections can make cross - comparison cumbersome. I suggest grouping figures by PFT (forest, grassland, cropland) with sub-panels for each site or model variant.**

Thank you for your helpful suggestion.

We agree that organizing the figures by plant functional type (PFT)—such as forest, grassland, and cropland—can improve clarity and facilitate more effective crosscomparisons. In response to your comment, we have revised the relevant figures accordingly, grouping them by PFT with sub-panels representing individual sites or model variants. This required a substantial amount of work, as it involved reprocessing the results and essentially rewriting this section of the manuscript. Nonetheless, we believe this reorganization enhances both the readability and interpretability of the results. We sincerely appreciate your constructive feedback.

Q6. The manuscript contains kind of repeated descriptions across all sessions. I recommend restructuring the whole manuscript thoroughly to avoid duplication. Meanwhile, the authors claim that all results are from validation, but without describing the split strategies.

Thank you for your valuable comment. In response, we have thoroughly restructured the manuscript to reduce redundancy and improve overall clarity. Repetitive descriptions across sections have been removed or streamlined to avoid duplication and enhance readability. Additionally, we have now clearly described the dataset split strategy in **Section 2.3.1** of the revised manuscript. Specifically, the pre-processed dataset was divided into training and testing sets using the Blocked Time Series Split strategy. Given the temporal dependency of the data, standard cross-validation is not suitable for time series analysis (Reichstein et al., 2019). Instead, a block-based and non-continuous split is applied to preserve the temporal structure. In this approach, the time series is partitioned into several non-overlapping continuous training blocks (e.g., 2003-2005, 2007-2009, 2011-2013, 2015-2017, 2019-2021), with independent years reserved as the validation set following each training block (e.g., 2006, 2010, 2014, 2018, 2022). This strategy ensures that the temporal order is maintained, preventing future data from leaking into the training process and thus avoiding invalid predictions. Additionally, the method incorporates validation over multiple periods, enabling the assessment of model generalization across different climate conditions, which is crucial for evaluating the model's robustness under varying environmental scenarios.

**Reference**

- Chang, X., Xing, Y., Gong, W., Yang, C., Guo, Z., Wang, D., Wang, J., Yang, H., Xue, G., Yang, S., 2023. Evaluating gross primary productivity over 9 ChinaFlux sites based on random forest regression models, remote sensing, and eddy covariance data. Sci. Total Environ. 875, 162601. https://doi.org/10.1016/j.scitotenv.2023.162601
- Hersbach, H., Bell, B., Berrisford, P., Hirahara, S., Horányi, A., Muñoz-Sabater, J., Nicolas, J., Peubey, C., Radu, R., Schepers, D., Simmons, A., Soci, C., Abdalla, S., Abellan, X., Balsamo, G., Bechtold, P., Biavati, G., Bidlot, J., Bonavita, M., De Chiara, G., Dahlgren, P., Dee, D., Diamantakis, M., Dragani, R., Flemming, J., Forbes, R., Fuentes, M., Geer, A., Haimberger, L., Healy, S., Hogan, R.J., Hólm, E., Janisková, M., Keeley, S., Laloyaux, P., Lopez, P., Lupu, C., Radnoti, G., de Rosnay, P., Rozum, I., Vamborg, F., Villaume, S., Thépaut, J.-N., 2020. The ERA5 global reanalysis. Q. J. R. Meteorol. Soc. 146, 1999–2049. https://doi.org/10.1002/qj.3803
- Reichstein, M., Camps-Valls, G., Stevens, B., Jung, M., Denzler, J., Carvalhais, N., Prabhat, 2019. Deep learning and process understanding for data-driven Earth system science. Nature 566, 195–204. https://doi.org/10.1038/s41586-019-0912-1

---

## Author Comment (AC6)

**Response to Anonymous Referee #3 (https://doi.org/10.5194/gmd-2024-169-**

**RC4)**

This is the review report for "FLAML version 2.3.3 model-based assessment of gross primary productivity at forest, grassland, and cropland ecosystem sites". The authors developed a FLAML modeling framework to predict vegetation gross primary productivity using hydro-meteorological variables and variables related to vegetation types and elevation. They focus on sites in China and provide detailed model performance in reproducing forest, grass, and crop sites. While the manuscript is structured, the authors need to check throughout the manuscript to ensure readability, e.g., the abbreviations. More importantly, I have some concerns about the modeling data input, cross-validation, and the selection of hydro-meteorological variables. Also, the evaluation of the absolute values of GPP can smooth out potentially poor performance during extreme situations. Therefore, a specific test for stress conditions and an evaluation of GPP anomalies are both highly recommended.

We are very grateful for your thoughtful and constructive comments, which have been instrumental in improving our manuscript. In response, we have carefully revised the manuscript by addressing each of the issues you pointed out. This process involved a substantial reorganization of the manuscript's structure and a comprehensive update of its content, resulting in extensive modifications throughout the text.

These revisions, we believe, have significantly enhanced both the clarity and scientific rigor of our work. Below, we provide detailed responses to each of your comments, explaining the corresponding changes made.

**Methodology**

Q1. GPP and RECO Partitioning: The manuscript should provide a clear description of the method used to partition GPP and RECO from NEE. Additionally, it is recommended to test different partitioning algorithms to assess their impact on the results.

Thank you for your insightful comment. Due to data upload inconsistencies, ER data were missing at several sites (DLG, LCA, XLG). To address this issue and ensure data consistency across all sites, we estimated ecosystem respiration (ER) using the Lloyd & Taylor equation (Reichstein et al., 2005; Lloyd and Taylor, 1994), which is a widely adopted method in flux data processing.

This approach distinguishes daytime and nighttime periods using shortwave radiation (Rg), with a threshold of  $10 \text{ W/m}^2$ . The temperature – response function derived from nighttime ER observations was then extrapolated to estimate daytime ER. This method is commonly used across many flux tower networks for separating Reco into GPP and ER components, and thus was adopted in our study to maintain methodological consistency. This has been clarified in **Section 2.2.1** of the revised manuscript.

We fully agree with your point that evaluating the impact of different partitioning algorithms on GPP estimation is valuable. However, in the context of this study, flux

partitioning serves as a preprocessing step rather than a primary research focus. A detailed comparison of flux partitioning methods would be more appropriate for a dedicated study, and we will consider exploring this direction in future work.

Q2. Train-Test Split Strategy: The procedure for splitting the dataset into training and validation sets needs to be described in greater detail. It is important to test whether the model maintains robustness during stress periods (e.g., droughts or heatwaves). Moreover, model performance should be evaluated not only in terms of seasonal GPP dynamics but also in reproducing GPP anomalies, which are crucial for capturing ecosystem responses beyond typical seasonal cycles.

Thank you for your valuable comments. We have addressed both of the concerns you raised through revisions and clarifications in the manuscript.

First, regarding the dataset split strategy, we have clearly described the methodology in **Section 2.3.1** of the revised manuscript. Specifically, the pre-processed dataset was divided into training and testing sets using the Blocked Time Series Split strategy. Given the temporal dependency of the data, standard cross-validation is not suitable for time series analysis (Reichstein et al., 2019). Instead, a block-based and noncontinuous split is applied to preserve the temporal structure. In this approach, the time series is partitioned into several non-overlapping continuous training blocks (e.g., 2003-2005, 2007-2009, 2011-2013, 2015-2017, 2019-2021), with independent years reserved as the validation set following each training block (e.g., 2006, 2010, 2014, 2018, 2022). This strategy ensures that the temporal order is maintained, preventing future data from leaking into the training process and thus avoiding invalid predictions. Additionally, the method incorporates validation over multiple periods, enabling the assessment of model generalization across different climate conditions, which is crucial for evaluating the model's robustness under varying environmental scenarios.

Second, regarding the evaluation of model performance under extreme environmental conditions, we have added corresponding analyses in **Section 3.2** of the revised manuscript. Numerous studies have shown that climate extremes—such as heatwaves, droughts, and high atmospheric vapor pressure deficit (VPD)—can significantly alter ecosystem functioning and reduce carbon uptake capacity (Frank et al., 2015; Reichstein et al., 2013). These events can suppress photosynthetic activity, increase respiration rates, and disrupt the carbon exchange balance between vegetation and the atmosphere. To evaluate the robustness and reliability of the FLAML-LUE model under such stress conditions, we examined model performance in simulating GPP during three types of climate extremes: high temperature, high VPD, and drought. By analyzing model accuracy and bias under these extreme scenarios, we aim to assess its applicability and limitations in challenging environmental settings.

Additionally, we acknowledge that the impacts of other extreme weather events and the ability of the model to reproduce GPP anomalies deserve further exploration, which we plan to address in future studies.

Thank you once again for your constructive feedback, which has helped us to improve the rigor and comprehensiveness of our study. Q3. Choice of Environmental Drivers: The exclusion of key hydrometeorological drivers such as precipitation, vapor pressure deficit (VPD), and soil moisture raise concerns. While LSWI and PDSI are included, they are indirect proxies and not physically direct controls of vegetation water uptake and stomatal regulation. The authors should justify this choice or consider incorporating more directly linked variables.

Thank you for your valuable suggestion. We fully agree that accurately representing hydrometeorological drivers is critical for modeling GPP and that variables such as precipitation, vapor pressure deficit (VPD), and soil moisture play important roles in regulating vegetation water uptake and stomatal conductance.

In our revised analysis, we have removed the PDSI dataset due to its coarse temporal resolution (monthly), which is inconsistent with the finer-scale (8-day or daily) datasets used in this study. Instead, we incorporated new variables that more directly and comprehensively capture vegetation moisture limitations from multiple ecological dimensions, based on both theoretical considerations and prior research (Chang et al., 2023):

- > Atmospheric moisture limitation: Relative humidity and precipitation
- > Vegetation-level moisture stress: LSWI and evaporative fraction (EF)
- Soil moisture limitation: Soil water content (SW)

We have updated the manuscript accordingly to clarify our variable selection rationale and better align with your suggestion.

**Specific points**

Q1. Line 101: The abbreviation "RFR" should be defined upon its first use for clarity. Thank you for your comment. We have revised the manuscript to define "RFR" (Random Forest Regressor) upon its first appearance to ensure clarity for the readers. Additionally, we have carefully reviewed the entire manuscript to identify and address any similar issues, and have made the necessary changes throughout the text.

Q2. Line 134: To promote transparency and reproducibility, the authors should provide a persistent identifier (e.g., DOI) for the datasets used, rather than referencing a general data repository that hosts multiple sources.

Thank you for your helpful suggestion. We have uploaded all datasets used in this study to Zenodo and provided a persistent identifier (DOI) for transparency and reproducibility. The data and code availability statement at the end of the manuscript has been updated accordingly: https://doi.org/10.5281/zenodo.14542880 (Laijie, 2024).

Q3. Line 178: ERA5-Land should not be categorized as remote sensing data. It is a reanalysis product based on assimilation of observations into a numerical model. Thank you for your valuable comments.

We have made corresponding revisions in the updated manuscript. Specifically,

**Section 2.2.3 (Line 181 - 192)** now reads as follows: "ERA5-Land (Hersbach et al., 2020) is a global high-resolution reanalysis dataset produced by the European Centre for Medium-Range Weather Forecasts (ECMWF) under the Copernicus Climate Change Service (C3S). It provides hourly land surface variables at a spatial resolution of 0.1°, generated using a dedicated land surface model driven by the ERA5 climate reanalysis. The dataset integrates advanced land surface modeling and data assimilation techniques, offering a wide range of variables such as air temperature, soil moisture, precipitation, and snow depth. In this study, site-specific variables including air temperature (T), soil water content (SW), precipitation (Pre), and leaf area index (LAI) were extracted from ERA5-Land. In addition, photosynthetically active radiation (PAR), evapotranspiration fraction (EF), VPD and relative humidity (RH) were calculated and derived from available ERA5-Land variables using GEE. "

**Q4. Line 195: The acronym "LSWI" should be spelled out in full the first time it appears.**

Thank you for your valuable comments. We have revised our manuscript as follows: Vegetation and water indices derived from MODIS data included the enhanced vegetation index (EVI), normalized difference vegetation index (NDVI), and land surface water index (LSWI), which were calculated using the formulas presented in Table 2 (Line 178).

Additionally, we have carefully reviewed the entire manuscript to identify and address any similar issues, and have made the necessary changes throughout the text.

Q5. Table 2: All abbreviations should be clearly defined either in the table caption or as a footnote to enhance readability.

Thank you for your valuable suggestion. We have revised Table 2 (Line 248) to include clear definitions of all abbreviations, which are now provided as footnotes to enhance clarity and readability.

Q6. Figure 3 (III): In model evaluation scatter plots, it is more intuitive to place observations on the x-axis and simulations on the y-axis, as this mirrors standard regression analysis practice.

Thank you for your valuable suggestion. We agree that placing observations on the xaxis and simulations on the y-axis provides a more intuitive interpretation and aligns with standard regression analysis practices. Following your recommendation, we have revised the scatter plot accordingly. In the updated version (now presented as **Figure 4**), we have also combined the three ecosystem types into a single figure to facilitate direct comparison across ecosystems.

**Figure 4.** Scatterplot of observed GPP vs. simulated GPP. Different colored dots represent different sites. Note: The simulated GPP values represent the mean of FLAML00 to FLAML25.

Q6. Figures 5/9/13: Do the reported biases account for seasonal differences in GPP variability (i.e., high variability in summer vs. low variability in winter)? Clarifying this would improve interpretation of model performance across seasons.

Thank you for your insightful comment. In our analysis, Figures 5/9/13 show the actual biases between the GPP simulations and observations across different sites and months. We acknowledge that the manuscript does not explicitly consider the seasonal variability in GPP. GPP tends to exhibit higher variability in summer and lower variability in winter, which may lead to higher GPP in summer and lower GPP in winter. In the revised manuscript, although we have included model evaluations under extreme climatic conditions, we have not specifically addressed the seasonal biases in the GPP simulations. Instead, we chose to use the PBias metric to provide an overall assessment of the model's performance across different land surface types. The PBias metric reflects the magnitude of simulation biases between sites, offering a more comprehensive evaluation of the model (Line 311 and Line 455).

**Q7. Figure 7: There is a noticeable underestimation of GPP in DLG (typical grasslands) and overestimation in DXG (alpine meadows). Can the authors explain potential causes for these systematic biases?**

Thank you for your valuable comment. In the revised manuscript (Line 441- 453), we used the PBias (%) metric to evaluate the simulation biases of different vegetation types. As shown in **Figure 5**, there is an underestimation of GPP at DLG (typical grasslands) and an overestimation at DXG (alpine meadows). These systematic biases can be attributed to differences in the biophysical characteristics and climatic

conditions of the two ecosystems.

For DLG, the grassland ecosystem typically exhibits high productivity under sufficient water availability, especially during the spring and summer growing seasons. If the model does not accurately represent the seasonal dynamics of water supply and demand, or the interaction between water availability and temperature, it may underestimate the actual GPP.

In contrast, GPP in alpine meadows like DXG is primarily constrained by low temperature and a short growing season. If the model does not fully capture these limitations—particularly under relatively cold conditions—it may overestimate the photosynthetic potential, resulting in an overestimation of GPP.

Q8. Figure 14: Are the farm ecosystems considered in the analysis purely rainfed, or do they include irrigated systems? This distinction is important for interpreting model results under water-limited conditions.

We sincerely thank the reviewer for the insightful comment regarding irrigation regimes at the cropland flux sites. Based on previous studies (Liu et al., 2023; Zhou et al., 2023; Zhang et al., 2023; Zhao et al., 2021), , the cropland ecosystems included in this study encompass both rainfed and irrigated systems. Specifically, SYA and JZA are rainfed single-cropping systems, where agricultural production primarily depends on natural precipitation. In contrast, GCA, LCA, and YCA are high-input, double-cropping systems located in intensively managed irrigated regions, where supplemental watering is essential during critical crop growth stages.

As stated in Section 4.1 of the revised manuscript (Lines 681 – 686), the current version of our model does not explicitly differentiate between the irrigation regimes of each site. Although we have identified the irrigation type for each location, this distinction has not yet been incorporated into the modeling framework. We fully recognize the pivotal role that irrigation plays in regulating GPP dynamics, particularly under water-limited conditions, and acknowledge that its exclusion may influence model performance and the scientific interpretation of results.

To address this limitation, we plan to integrate satellite-derived irrigation indicators in future studies—specifically, soil moisture anomalies from the Soil Moisture Active Passive (SMAP) mission and temporal patterns of the Normalized Difference Water Index (NDWI). Incorporating these indicators will enhance the model's ability to represent irrigation effects and more accurately capture the dynamic variability of carbon fluxes in agricultural ecosystems.

**Reference**

- Chang, X., Xing, Y., Gong, W., Yang, C., Guo, Z., Wang, D., Wang, J., Yang, H., Xue, G., Yang, S., 2023. Evaluating gross primary productivity over 9 ChinaFlux sites based on random forest regression models, remote sensing, and eddy covariance data. Sci. Total Environ. 875, 162601. https://doi.org/10.1016/j.scitotenv.2023.162601
- Frank, Dorothea, Reichstein, M., Bahn, M., Thonicke, K., Frank, David, Mahecha, M.D., Smith, P., van der Velde, M., Vicca, S., Babst, F., Beer, C., Buchmann, N., Canadell, J.G., Ciais, P.,

Cramer, W., Ibrom, A., Miglietta, F., Poulter, B., Rammig, A., Seneviratne, S.I., Walz, A., Wattenbach, M., Zavala, M.A., Zscheischler, J., 2015. Effects of climate extremes on the terrestrial carbon cycle: concepts, processes and potential future impacts. Glob. Change Biol. 21, 2861–2880. https://doi.org/10.1111/gcb.12916

- Hersbach, H., Bell, B., Berrisford, P., Hirahara, S., Horányi, A., Muñoz-Sabater, J., Nicolas, J., Peubey, C., Radu, R., Schepers, D., Simmons, A., Soci, C., Abdalla, S., Abellan, X., Balsamo, G., Bechtold, P., Biavati, G., Bidlot, J., Bonavita, M., De Chiara, G., Dahlgren, P., Dee, D., Diamantakis, M., Dragani, R., Flemming, J., Forbes, R., Fuentes, M., Geer, A., Haimberger, L., Healy, S., Hogan, R.J., Hólm, E., Janisková, M., Keeley, S., Laloyaux, P., Lopez, P., Lupu, C., Radnoti, G., de Rosnay, P., Rozum, I., Vamborg, F., Villaume, S., Thépaut, J.-N., 2020. The ERA5 global reanalysis. Q. J. R. Meteorol. Soc. 146, 1999–2049. https://doi.org/10.1002/qj.3803
- Lloyd, J., Taylor, J.A., 1994. On the Temperature Dependence of Soil Respiration. Funct. Ecol. 8, 315–323. https://doi.org/10.2307/2389824
- Reichstein, M., Bahn, M., Ciais, P., Frank, D., Mahecha, M.D., Seneviratne, S.I., Zscheischler, J., Beer, C., Buchmann, N., Frank, D.C., Papale, D., Rammig, A., Smith, P., Thonicke, K., van der Velde, M., Vicca, S., Walz, A., Wattenbach, M., 2013. Climate extremes and the carbon cycle. Nature 500, 287–295. https://doi.org/10.1038/nature12350
- Reichstein, M., Camps-Valls, G., Stevens, B., Jung, M., Denzler, J., Carvalhais, N., Prabhat, 2019. Deep learning and process understanding for data-driven Earth system science. Nature 566, 195–204. https://doi.org/10.1038/s41586-019-0912-1
- Reichstein, M., Falge, E., Baldocchi, D., Papale, D., Aubinet, M., Berbigier, P., Bernhofer, C., Buchmann, N., Gilmanov, T., Granier, A., Grünwald, T., Havránková, K., Ilvesniemi, H., Janous, D., Knohl, A., Laurila, T., Lohila, A., Loustau, D., Matteucci, G., Meyers, T., Miglietta, F., Ourcival, J.-M., Pumpanen, J., Rambal, S., Rotenberg, E., Sanz, M., Tenhunen, J., Seufert, G., Vaccari, F., Vesala, T., Yakir, D., Valentini, R., 2005. On the separation of net ecosystem exchange into assimilation and ecosystem respiration: review and improved algorithm. Glob. Change Biol. 11, 1424–1439. https://doi.org/10.1111/j.1365-2486.2005.001002.x
- Liu, F., Shen, Y.J., Cao, J.S., Zhang, Y.C., 2023. A dataset of water, heat, and carbon fluxes over the winter wheatsummer maize croplands in Luancheng during 2013-2017. China Scientific Data (in Chinese), 8, 36–45.
- Zhou, L., Geng, J.J., Zhou, G.S., Zhang, S., Wu, Y.X., 2023. A dataset of carbon and water fluxes in the winter wheat and summer maize rotation ecosystem at Gucheng Station (2020– 2022). China Scientific Data (in Chinese), 8, 68–77.
- Zhang, S, Zhou, L., Zhou, G.S., Jia, Q.Y., Li, R.P., Wang, Y., 2023. A dataset of carbon and water flux observations in the agricultural ecosystem of spring maize in Jinzhou (2005–2014). China Scientific Data (in Chinese), 8, 148–159.
- Zhao, F.H., Li, F.D., Zhan, C.S., Zhang, L.M., Chen, Z., 2021. A carbon and water fluxes dataset of the farmland ecosystem of winter wheat and summer maize in Yucheng (2003–2010). China Scientific Data (in Chinese), 6, 222–228.

---

## Author Response (AR2)

**Response to Topic editor**

Q1. Lack of Explanation of FLAML Model Structures: From the current description, it is only mentioned that FLAML uses an automated framework (AutoML) for building the models, but there is no description on the type of models that are tested, and the type of models that were selected by the procedure. The article only mentions the variables that were selected, but not the type of model found by the AutoML. I believe this is an important piece of information missing from the entire article. Right now, the article only focuses on evaluating model performance through a set of metrics, but there are no new insights regarding the value of FLAML in selecting certain types (structures) of models. I recommend the authors to better describe the type of models that are tested by FLAML in section 2.3.2. The focus should be on the structural characteristics of the model, not on the predictive variables, which are well explained in section 2.3.3. and do not need further elaboration.

We sincerely appreciate your valuable comments. In response to the two main concerns raised, we have revised the manuscript accordingly and provide the detailed replies below.

Firstly, the lack of structural description of the models tested by FLAML. Accordingly, we have revised Section 2.3.2 to provide more detailed explanations of the candidate model types used by FLAML, focusing on their structural characteristics and differences in learning strategy, randomness, and ensemble behavior. The below changes have been made to Section 2.3.2 (Lines 265–287):

"For our regression tasks, AutoML was configured with the "auto" option for the estimator list, focusing on optimizing the $R^2$ metric and using a time budget of 120 seconds per run. Under this "auto" setting, FLAML explores a variety of built-in regression estimators, including:

1. LightGBM (Ke et al., 2017): a histogram-based gradient boosting method designed for speed and scalability;
2. XGBoost (Chen and Guestrin, 2016): a regularized gradient boosting framework known for its robustness and accuracy;
3. CatBoost (Prokhorenkova et al., 2018): efficiently handles categorical features and reduces overfitting via ordered boosting;
4. Random Forest (Breiman, 2001): an ensemble method utilizing bootstrap aggregation of decision trees;
5. Extra Trees (Geurts et al., 2006): enhances randomness in split point selection for tree construction;
6. Histogram-based Gradient Boosting (Brownlee, 2020), accelerate training through feature binning;
7. K-Nearest Neighbors (Cover and Hart, 1967): a non-parametric distance-based algorithm relying on local data density;
8. Transformer models (Vaswani et al., 2023), deep learning architectures leveraging self-attention mechanisms, adapted here for structured data regression.
   Collectively, these estimators span a broad algorithmic spectrum, including

ensemble learning, distance-based methods, and neural networks, enabling FLAML to automatically identify the optimal model architecture for the dataset and objective."

Secondly, to address the issue that the manuscript previously did not report which models AutoML ultimately selected, we have added new content in Section 3.1 (Lines 351–360):

"To evaluate the model performance at the site level, the accuracy of the 18 FLAML-LUE models was assessed using test datasets from individual flux tower sites. The algorithms selected by each FLAML-LUE model are listed in **Table S1**. Notably, the Extra-Trees algorithm was most frequently chosen as the best-performing model. Extra Trees is an ensemble method that constructs multiple unpruned decision trees and introduces high randomness in both feature and threshold selection, which enhances generalization and reduces overfitting, particularly in noisy or high-dimensional datasets. The consistent selection of Extra Trees suggests that FLAML tends to favor models with higher stochasticity and ensemble structures under the given data and computational constraints."

We believe these revisions adequately address the reviewers' concerns by clarifying the range of candidate models tested and by highlighting FLAML's model selection process in our study.

**Table S1**
Optimal algorithms of the FLAML-LUE Model for GPP Simulation Across Different Temporal Scales and Predictor Combinations.

| FLAML | Daily | 8-day | 16-day | Monthly |
|---|---|---|---|---|
| FLAML00 | extra_tree | extra_tree | extra_tree | extra_tree |
| FLAML01 | extra_tree | extra_tree | extra_tree | extra_tree |
| FLAML02 | extra_tree | extra_tree | extra_tree | extra_tree |
| FLAML03 | extra_tree | extra_tree | xgb_limitdepth | extra_tree |
| FLAML04 | rf | rf | extra_tree | extra_tree |
| FLAML05 | extra_tree | extra_tree | extra_tree | extra_tree |
| FLAML10 | extra_tree | extra_tree | extra_tree | extra_tree |
| FLAML11 | extra_tree | rf | extra_tree | extra_tree |
| FLAML12 | extra_tree | extra_tree | extra_tree | extra_tree |
| FLAML13 | extra_tree | extra_tree | extra_tree | extra_tree |
| FLAML14 | lgbm | extra_tree | extra_tree | extra_tree |
| FLAML15 | extra_tree | extra_tree | extra_tree | extra_tree |
| FLAML20 | extra_tree | extra_tree | extra_tree | extra_tree |
| FLAML21 | lgbm | extra_tree | extra_tree | extra_tree |
| FLAML22 | extra_tree | extra_tree | extra_tree | extra_tree |
| FLAML23 | extra_tree | extra_tree | extra_tree | extra_tree |
| FLAML24 | rf | rf | rf | extra_tree |
| FLAML25 | extra_tree | extra_tree | extra_tree | extra_tree |

Q2. Insufficient Discussion on the Value of AutoML (FLAML): The discussion section should include something about the structure of the selected models and why the AutoML procedure has advantages over other ML methods.

Thank you for your valuable suggestion. In the revised Discussion section, we have further expanded on the structure of the models selected by FLAML and emphasized the advantages of using AutoML. Specifically, we highlight FLAML's ability in conducting automatic search across a diverse set of model families and configurations, adaptively selecting optimal models based on their performance metrics, and minimizing the need for manual trial-and-error. These features make FLAML a practical and efficient tool for environmental modeling tasks. Specifically, we have added the following text to the revised manuscript (Lines 788–810):

"In this study, FLAML (Wang et al., 2021) selected the Extra Trees algorithm as the best-performing model for GPP simulation in China. Extra Trees is an ensemble learning method that builds multiple unpruned decision trees and incorporates randomization in features selection and split thresholds determination. Compared to traditional decision tree ensembles such as Random Forests, Extra Trees typically achieves minimal variance while maintaining low bias, which makes it particularly well-suited for complex, high-dimensional datasets (Geurts et al., 2006).

The adoption of FLAML provides several significant advantages. First, it automates the model selection and hyperparameter tuning process, eliminating the need for extensive manual trial-and-error and reducing reliance on domain expertise (Nakano and Liu, 2025; Wang et al., 2022). Instead of manually evaluating various algorithms and their configurations, FLAML efficiently explores a broad search space and identifies the most appropriate model for the dataset.

Moreover, FLAML employ a cost-aware hyperparameter optimization strategy, enabling it to find high-performing models with relatively low computational cost (Zhang et al., 2023; Wang et al., 2021). This feature is particularly advantageous in scenarios with limited computational resources or the need for rapid prototyping.

Compared to conventional machine learning workflows, FLAML significantly reduces human bias in model selection, improves reproducibility, and lowers the barrier to applying advanced modeling techniques (He et al., 2021). Overall, the use of FLAML in this study not only improved model performance but also streamlined the modeling process, supporting its broader applicability in ecological and climate-related research."

Q3. Inadequate Clarification on LUE Model Use: Reviewer 2 in Q2 made an important comment on the use of LUE in the text but not using an explicit LUE model. The answer you provided is a good answer, but I didn't see an explicit text in the manuscript on this topic. In your response you mentioned that lines 122 and 272 include this information, but the text in the revised version is too vague. Please include a similar text as what you provided in your answer to reviewer 2 in section 2.3.3.

Thank you for pointing this out. In response to your comment, we have revised Section 2.3.3 of the manuscript to provide a clearer and more explicit explanation of

how LUE theory is incorporated into our modeling approach. Specifically, we have added the following text (Lines 289–299):

"Eighteen FLAML-LUE model variations were constructed for all sites by combining different permutations of six input factor groups, as described in Eq. (3) and detailed in Table 3. Technically, the term "FLAML-LUE" does not refer to a direct implementation of a mechanistic LUE model. Instead, it reflects a hybrid modeling strategy, through which we incorporate key explanatory variables that originate from LUE theory—such as fPAR, light-use efficiency modifiers, and environmental stress indicators (e.g., VPD, temperature, and water stress indices)—into an automated machine learning framework (FLAML). These variables capture the main drivers of vegetation productivity in traditional LUE models. Their integration enables FLAML to build models that are both ecologically grounded and predictive, effectively balancing model interpretability and accuracy."

We sincerely appreciate the editor's constructive suggestions, which have substantially improved the clarity and rigor of our manuscript. We hope that the revised version now meets the expectations for publication.